# GENERATING MODEL PARAMETERS FOR CONTROLLING: PARAMETER DIFFUSION FOR CONTROLLABLE MULTI-TASK RECOMMENDATION

## ABSTRACT

Commercial recommender systems face the challenge that task requirements from platforms or users often change dynamically (e.g., varying preferences for accuracy or diversity). Ideally, the model should be re-trained after resetting a new objective function, adapting to these changes in task requirements. However, in practice, the high computational costs associated with retraining make this process impractical for models already deployed to online environments. This raises a new challenging problem: how to efficiently adapt the learning model to different task requirements by controlling model parameters after deployment, without the need for retraining. To address this issue, we propose a novel controllable learning approach via **Pa**rameter **Di**ffusion for controllable multi-task **Rec**ommendation (**PaDiRec**), which allows the customization and adaptation of recommendation model parameters to new task requirements without retraining. Specifically, we first obtain the optimized model parameters through adapter tunning based on the feasible task requirements. Then, we utilize the diffusion model as a parameter generator, employing classifier-free guidance in conditional training to learn the distribution of optimized model parameters under various task requirements. Finally, the diffusion model is applied to effectively generate model parameters in a test-time adaptation manner given task requirements. As a model-agnostic approach, PaDiRec can leverage existing recommendation models as backbones to enhance their controllability. Extensive experiments on public datasets and a dataset from a commercial app, indicate that PaDiRec can effectively enhance controllability through efficient model parameter generation. The code is released at https://anonymous.4open.science/r/PaDiRec-DD13e.

## 1 INTRODUCTION

Traditional recommender systems are usually designed to improve accuracy by analyzing user behaviors and contextual data to uncover users' potential interests and preferences (Kang & McAuley, 2018; Hidasi et al., 2016). Nowadays, recommendation models place greater emphasis on multiple important aspects of the recommended results (also called multi-task recommendation), such as diversity (Xia et al., 2017), fairness (Oosterhuis, 2021), etc. Existing multi-task recommendation models are typically static (Zhang & Yang, 2021; Sener & Koltun, 2018), meaning that the preference weights for each aspect (e.g., accuracy or diversity) are predefined and fixed during both training and testing. Once the *static* preference weights are determined, the training process can employ various optimization algorithms to find the optimal solution.

However, in practical scenarios, the preference weights for different aspects often *change dynamically* across both context and time. From a commercial perspective, different application scenarios may require varying preference weights for different performance aspects of the recommendation model to meet specific business needs. For instance, the checkout page emphasizes product diversity, while the product detail page prioritizes accuracy by recommending similar items. From the users' perspective, different user groups may have distinct preferences, and even the same users may have changing information needs over time. For example, a user may prefer highly accurate recommendations when browsing a specific item category, but over time, such precision might diminish their interest, prompting a preference for more diverse categories. To address the above

dynamic information needs of users or platforms, this paper focuses on enhancing the controllability of recommendation models at test time, specifically in the context of controllable multi-task recommendation.

Traditional multi-task learning approaches face challenges in addressing the issue of dynamically changing preference weights. More specifically, when preference weights change, they require resetting the objective function, re-training the recommendation model based on the new objective, and then redeploying the updated model. However, while this approach enables the integration of various optimization methods, the retraining process is highly time- and resource-intensive, rendering it impractical — especially since rapid response time is critical during online recommendation phases. For instance, during promotional events, commercial stores often require real-time flow control to adjust their recommendation strategies, with changes ideally implemented immediately. Several studies have recognized the importance of dynamically adjusting models based on changing preferences. Wortsman et al. (2022) used simple parameter merging across multiple task-specific models, and Chen et al. (2023) employed discriminative models to generate parameters for multitask re-ranking problem. While they reduce response time and aim to enhance control over the model, they struggle with approximating the optimal model (which we assume can be achieved through retraining with given preference weights), potentially leading to suboptimal solutions. To achieve both efficient test-time adaptation to changing preferences and preserve the approximate optimal performance that retraining offers, we leverage the strengths of diffusion models in generating high-performance model parameters (Schürholt et al., 2022; Knyazev et al., 2021; Wang et al., 2024) for recommendation model. Additionally, we utilize conditional control (Ho & Salimans, 2022) to ensure controllability at test-time with changing preference weights as conditions.

In this work, we propose a novel parameter generation approach for controllable multi-task recommendation by leveraging a generative model to efficiently generate task-specific model parameters at test time based on varying task requirements (i.e., the preference weights for different performance metrics), effectively addressing the challenges posed by rapidly changing requirements and the high cost of retraining models. The proposed approach, termed PaDiRec, begins by formulating an objective function aligned with task-specific preference weights, and through advanced optimization techniques, we fine-tune model parameters using adapter tuning. We then train a diffusion model to learn the conditional distribution of these optimized adapter parameters under various task requirements, where the classfier-free guidance training strategy is employed to perform conditional training. Once trained, during online testing, the diffusion model can generate task-specific adapter parameters with the task requirement as condition, which can be integrated with different sequential recommendation backbones to produce recommendation lists that meet the specified requirements. Additionally, PaDiRec is both model-agnostic and algorithm-agnostic, making it flexible and compatible with various recommendation models and optimization strategies. We summarize our contributions as follows:

- We formally define the problem of controllable multi-task recommendation (CMTR), which focuses on the model's ability to adapt to dynamic changes in preferences for different metrics during online testing.
- We present PaDiRec, a diffusion model-based approach that generates model parameters conditioned on task-specific preference weights, providing enhanced control and flexibility by controlling model parameters in multi-task learning settings.
- Extensive experiments on two public datasets and an industrial dataset demonstrate that PaDiRec achieves superior performance towards controllability of multi-task recommendation while retaining recommendation performances.

## 2 PROBLEM FORMULATION AND ANALYSES

Given a user $u \in \mathcal{U}$ and a set of candidate items $\mathcal{C} = \{c_k\}_{k=1}^{|\mathcal{C}|}$ where $|\mathcal{C}|$ denotes the total number of candidate items. the historical interaction sequence of user $u$ of length $h$ is denoted by $S_u = \{c_1^u, c_2^u, \ldots, c_h^u\}$ (also called user history), where $c_k^u \in \mathcal{C}, k \in \{1, 2, \ldots, h\}$. For a **recommendation task** $i \in \{1, 2, \ldots, N\}$, a recommender system aims to find the following item list $L_i^*$ among all possible lists $\{L\}$ composed by candidate items from $\mathcal{C}$:

$$L_i^* = \arg\max_L R_i(L \mid S^u, \mathcal{C}), \qquad (1)$$

where $R_i$ denotes the reward function corresponding to task $i$, which evaluates the recommender system's performance with respect to task $i$. More specifically, modern recommender systems often evaluate performance from multiple perspectives, the reward function in Eq. (1) for task $i$ can be expressed as the following linear combination of $p$ utility functions $\{U_j\}_{j=1}^p$:

$$R_i(L(S_u, \mathcal{C})) = \sum_{j=1}^p w_i^j \, U_j(L \mid S_u, \mathcal{C}), \qquad (2)$$

which allows task $i$ to be quantified by a set of **preference weights** $\boldsymbol{w}_i = \{w_i^j\}_{j=1}^p \in \mathcal{W}$ for the various utilities, where $\mathcal{W}$ denotes the preference weight space that is a simplex.

Then, we can provide the definition of **controllable multi-task recommendation (CMTR)**. The goal of CMTR is to find a recommendation model $f_{\boldsymbol{\theta}}$, parameterized by $\boldsymbol{\theta} \in \Theta$, such that the item lists output during test time, $L = f_{\boldsymbol{\theta}}(S_u, \mathcal{C})$, can adapt to changes in tasks (i.e., adapt to variations in the corresponding preference weights in Eq. (2)). As an example, after the recommendation model $f_{\boldsymbol{\theta}}$ is deployed, when the preference weights for different utilities (e.g., accuracy and diversity) need to shift from $\boldsymbol{w}_i = \{w_i^j\}_{j=1}^p$ (i.e., task $i$) to $\boldsymbol{w}_k = \{w_k^j\}_{j=1}^p$ (i.e., task $k$) based on user or platform requirements, we say that the recommendation model $f_{\boldsymbol{\theta}}$ is **controllable** if it can ensure that its reward remains at a high level regardless of how the preference weights change. Ideally, to accommodate changes in tasks, we could retrain the recommendation model after receiving new preference weights to update its parameters, resulting in $f_{\tilde{\boldsymbol{\theta}}}$ that maintains a high reward. However, for an already deployed model, the time required for retraining is impractical and unacceptable. Another straightforward method would be to store $N$ sets of task-specific parameters corresponding to the preference weights for $N$ tasks at the time of deployment, and load them when a new task arises at test time. However, when considering a continuous preference weight space where the number of tasks $N$ tends to infinity (i.e., a continuous task space), this discrete method becomes impractical due to storage limitations and cannot accommodate fine-grained or continuous task variations.

To efficiently and effectively adapt to changes in tasks, this paper focuses on controlling the model parameters $\boldsymbol{\theta}$ of the recommendation model $f_{\boldsymbol{\theta}}$ to accommodate the varying preference weights of new tasks. More specifically, we treat the preference weights as variables and model the relationship between the preference weight space $\mathcal{W}$ and the model parameter space $\Theta$ during training, transforming the time- and resource-intensive retraining problem at test time into an efficient inference problem. Formally, we aim to find a function $g_{\boldsymbol{\xi}} : \mathcal{W} \to \Theta$ (where $\boldsymbol{\xi}$ denotes the parameter of $g$) that generates model parameters capable of achieving a high reward given the new preference weights $\boldsymbol{w}_k$ for any task $k$ at test time:

$$R_k(L(S_u, \mathcal{C})) = \sum_{j=1}^p w_k^j \, U_j(f_{\boldsymbol{\theta}_k}(S_u, \mathcal{C}) \mid \boldsymbol{\theta}_k = g_{\boldsymbol{\xi}}(\boldsymbol{w}_k)). \qquad (3)$$

In contrast to traditional multi-task recommendation (MTR), which focuses only on *fixed* preference weights for different utilities, our defined CMTR emphasizes how the model adapts to *dynamic* changes in preference weights after deployment. This shift means that in traditional MTR, each task corresponds to a *single* utility, whereas in CMTR, each task is associated with *multiple* utilities combined through a linear weighting, with combination coefficients determined by a set of task-specific preference weights. As a result, CMTR places greater emphasis on test-time adaption to handle dynamic task requirements, introducing new challenges for CMTR model training and construction compared to MTR.

## 3 PADIREC: THE PROPOSED APPROACH

In this section, we provide a detailed description of the proposed approach, **PaDiRec**. PaDiRec utilizes a conditional generative framework designed to directly learn from the optimized parameters of recommendation models tailored to specific tasks. This pre-training process enables the generation of new model parameters based on specified preference weights at test time.

### 3.1 ALGORITHM OVERVIEW

As shown in Figure 1, we provide an illustrative overview of the proposed PaDiRec, which contains the following three phases. (1) *Preparation of adapters*: the left part in Figure 1 shows the training

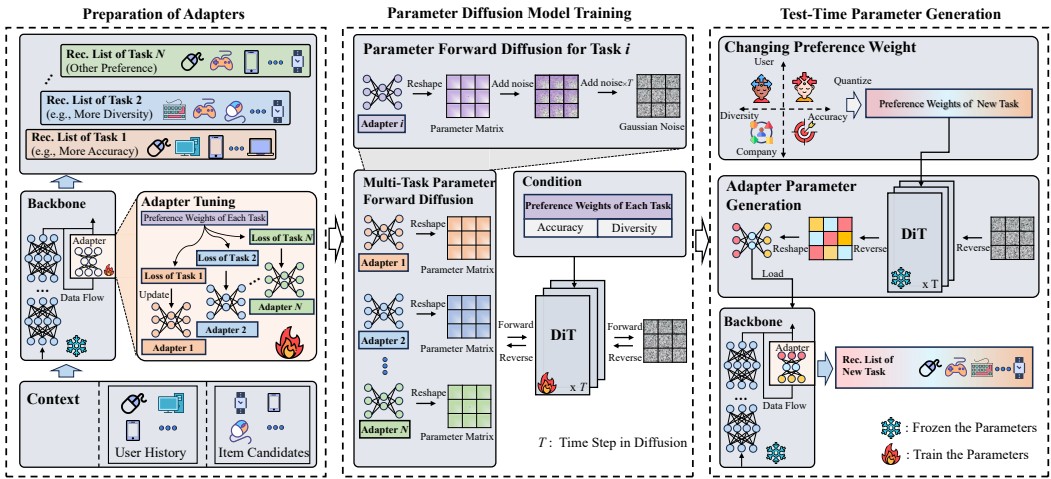

Figure 1: An overview of the proposed PaDiRec. Details are shown in Sec. 3.1

process of the recommendation model, from which we can obtain a collection of optimized adapter parameters for feasible specific task by sampling the preference weights. Note that we focus on two utilities: accuracy and diversity. As defined in Eq. (3), each task is represented by a set of preference weights for these two utilities. (2) *Parameter diffusion model training*: the middle part in Figure 1 illustrates the conditional training procedure of the generative model $g_{\boldsymbol{\xi}}$ (i.e., DiT) with the optimized adapter parameters as initial data and the corresponding preference weights as condition, thus generating meaningful adapter parameters from Gaussian noise given preference weights. (3) *Test-time parameter generation*: the right part in Figure 1 shows how we utilize the trained DiT model during the test phase to adapt to dynamically changing task requirements (i.e., preference weights for diversity and accuracy). First, we quantify these task requirements as preference weights. Next, we employ the trained DiT model to generate adapter parameters in real time, using these preference weights as inputs, which are then combined with the backbone to directly support the recommendation task. In the following subsections, we elaborate on the details of these phases.

## 3.2 PREPARATION OF ADAPTERS

Our goal is to construct the parameters of optimized recommendation models under different preference weights to prepare data for the generative model. Thus, this section is organized into three parts: the structure of the recommendation model, the construction of task-specific objective functions, and the tuning process for the recommendation model parameters.

**Model structure.** As shown in the left module of Figure 1, sequential recommendation models take user history and candidate items as input. Guided by the objective function (i.e., loss function), the model learns the underlying relationships within the user history, ultimately generating a recommendation list (i.e., Rec. List) from the candidate items. Existing recommender systems based on deep neural networks can be quite large, and making significant invasive modifications typically requires retraining the entire model, which is often prohibitively expensive in industrial applications. To address this, our approach introduces an adapter module, which can be seamlessly integrated into existing sequential recommendation models. Specifically, we incorporate the adapter using a residual connection, attaching it to the last layer of the backbone model. In this setup, the backbone is set to retain the original recommendation capabilities, while the adapter is responsible for adapting to specific tasks.

**Objective function construction.** To obtain the optimized task-specific adapter parameters under CMTR setting (as shown in Sec. 2 ), we first focus on the construction of loss functions based on different preference weights of each task. Specifically, we directly convert the reward maximization problem (reward defined in Eq. (2)) into a loss minimization problem. Given a specific set of preference weights $\boldsymbol{w}_i = \{w_i^j\}_{j=1}^p \in \mathcal{W}$, which represent preference weight for the $j$-th utility in the requirement of task $i$. Here, we focus on two utilities including diversity loss $\ell_{\text{diversity}}$ and accuracy

loss $\ell_{\text{accuracy}}$ in each task (i.e., $p = 2$). Thus, the total loss function for task $i$ is

$$\ell_i = w_i^1 \ell_{\text{accuracy}} + w_i^2 \ell_{\text{diversity}}. \tag{4}$$

**Adapter tuning.** Based on above total loss function, we decompose the recommendation model parameters $\boldsymbol{\theta}$ into two components: task-specific adapter parameters, denoted as $\boldsymbol{\theta}_a$ and task-independent backbone parameters, denoted as $\boldsymbol{\theta}_b$. Accordingly, optimizing the model is divided into two phases. The *first phase* focuses on optimizing the backbone parameters $\boldsymbol{\theta}_b$, which uses the standard BCE loss to train the backbone model thus preserving the original recommendation accuracy. The *second phase* is about the optimization of the task-specific adapter parameters $\boldsymbol{\theta}_a$, which aims at improving the system's adaptability to different tasks. During the second phase, the backbone parameters are frozen to prevent them from being tailored to any specific task, whereas the adapter is trainable. More specifically, in the second phase, we train the task-specific adapter parameters based on two loss functions as in Eq. (4), one for accuracy and one for diversity. For the accuracy loss $\ell_{\text{accuracy}}$, we continue to use BCE as the loss function to guide the model toward accuracy. For the diversity loss $\ell_{\text{diversity}}$, inspired by Yan et al. (2021), we apply a differentiable smoothing of the $\alpha$-DCG metric and adapt it to the recommendation setting. Consider $|\mathcal{C}|$ candidate items and $|\mathcal{M}|$ categories, where each item may cover 0 to $|\mathcal{M}|$ categories. The category labels are denoted as $y_{k,l}$: $y_{k,l} = 1$ if item $k$ covers category $m$, and $y_{k,l} = 0$ otherwise, where $k \in \{0, \dots, |\mathcal{C}| - 1\}$, $l \in \{0, \dots, |\mathcal{M}| - 1\}$. Based on the $\alpha$-DCG, we design a differentiable diversity loss function:

$$\ell_{\text{diversity}} = -\sum_{k=1}^{|\mathcal{C}|} \sum_{l=1}^{|\mathcal{M}|} \frac{y_{k,l}(1-\alpha)C_{k,l}}{\log_2(1 + \text{Rank}_k)}, \tag{5}$$

where $\alpha$ is a hyper parameter between 0 and 1, $\text{Rank}_k$ is the soft rank of the item $k$, and $C_{k,l}$ is the number of times the category $l$ being covered by items prior to the soft rank $\text{Rank}_k$. That is:

$$\text{Rank}_k = 1 + \sum_{j \neq k} \text{sigmoid}\left((s_j - s_k)/T\right), \quad C_{k,l} = \sum_{j \neq k} y_{j,l} \cdot \text{sigmoid}\left((s_j - s_k)/T\right), \tag{6}$$

where $s_k$ denotes the relevance score of the $k$-th candidate item output by the model. For task $i$, we denote $\boldsymbol{\theta}_i$ as the model parameters including task-specific adapter parameters $\boldsymbol{\theta}_i^a$ and fixed backbone parameters $\boldsymbol{\theta}_i^b$. Overall, based on the total loss in Eq. (4), the task-specific optimization process of $\boldsymbol{\theta}_i^a$ for task $i$ can be formulated as follows:

$$\boldsymbol{\theta}_i^a = \arg\min_{\boldsymbol{\theta}_i^a} \ w_i^1 \ell_{\text{accuracy}} + w_i^2 \ell_{\text{diversity}}, \tag{7}$$

where $\boldsymbol{w}_i = \{w_i^1, w_i^2\} \in \mathcal{W}$ is sampled from $[0, 1]$. We employ the standard Adam optimizer to optimize these parameters. Then we transform the parameters of each task-specific adapter into a matrix-based format and these optimized parameters serve as the ground truth for the subsequent generative model training process.

### 3.3 PARAMETER DIFFUSION MODEL TRAINING

The optimized adapter parameters and corresponding preference weights obtained from Sec. 3.2 are used as the training data for the diffusion model. We employ a generative model $g_{\boldsymbol{\xi}}$ parameterized by $\boldsymbol{\xi}$ to learn the process of generating model parameters. Specifically, $g_{\boldsymbol{\xi}}$ is applied to predict the conditional distribution of the adapter parameter matrices $p_{g_{\boldsymbol{\xi}}}(\boldsymbol{\theta}_i^a | \boldsymbol{w}_i)$ given the preference weights $\boldsymbol{w}_i$, where $i$ corresponds to the task $i$. We adopt diffusion models (Ho et al., 2020) as our generative model due to its efficacy in various generation tasks (Li et al., 2022; Ho et al., 2022a; Vignac et al., 2023) and its superior performance on multi-modal conditional generation (Bao et al., 2023; Nichol et al., 2022; Saharia et al., 2022). We train the diffusion model to sample parameters by gradually denoising the optimized adapter parameter matrix from the Gaussian noise. This process is intuitively reasonable as it intriguingly mirrors the optimization journey from random initialization which is a well-established practice in existing optimizers like Adam. For task $i$, our denoising model takes two parts as the input: a noise-corrupted adapter parameter matrix $\boldsymbol{\theta}_{i,t}^a$, and a set of preference weights $\boldsymbol{w}_i$, with $t$ representing the step in the forward diffusion process. The training objective is as follows:

$$\ell_{\text{diff}} = \mathbb{E}_{\boldsymbol{\theta}_{i,0}^a, \epsilon \sim \mathcal{N}(0,1), t} \left[ \left\| \epsilon - \epsilon_{\boldsymbol{\xi}}(\boldsymbol{\theta}_{i,t}^a, \boldsymbol{w}_i, t) \right\|^2 \right], \tag{8}$$

where $\epsilon$ denotes the noise to obtain $\boldsymbol{\theta}^{\mathrm{a}}_{i,t}$ from $\boldsymbol{\theta}^{\mathrm{a}}_{i,0}$, and the denoising model $\epsilon(\cdot)$ is the main part of the generative model $g_{\boldsymbol{\xi}}$. We assume that the parameters of $g_{\boldsymbol{\xi}}$ primarily originate from the denoising model. For simplicity, we denote the denoising model as $\epsilon_{\boldsymbol{\xi}}$. To conduct condition training in a classifier-free guidance manner (Ho & Salimans, 2022), we use the denoising model to serve as both the conditional and unconditional model by simply inputting a null token $\varnothing$ as the condition (i.e., preference weights $\boldsymbol{w}_i$) for the unconditional model, i.e. $\epsilon_{\boldsymbol{\xi}}(\boldsymbol{\theta}^{\mathrm{a}}_{i,t},t) = \epsilon_{\boldsymbol{\xi}}(\boldsymbol{\theta}^{\mathrm{a}}_{i,t}, \boldsymbol{w}_i = \varnothing, t)$. The probability of setting $\boldsymbol{w}_i$ to $\varnothing$ is denoted as $p_{\mathrm{uncond}}$ and is configured as a hyperparameter. Alg. 1 in Appendix illustrates the detailed procedure.

## 3.4 TEST-TIME PARAMETER GENERATION

After the diffusion is trained, we can generate the parameters $\boldsymbol{\theta}^{\mathrm{a}}_{n,0}$ by querying $g_{\boldsymbol{\xi}}$ with a new set of preference weights $\boldsymbol{w}_n$, specifying the desired preference weights for accuracy and diversity of new task $n$. Then the generated adapter parameter $\boldsymbol{\theta}^{\mathrm{a}}_{n,0}$ for that new task is directly loaded into the adapter, which is connected to the backbone. This forms a new customized recommendation model that responds to the preference weights of the new task. The generation is an iterative sampling process from step $t = T$ to $t = 0$, which denoises the Gaussian noise into meaningful parameters taking specific preference weights as the condition. The generation process is formulated as follows:

$$\tilde{\epsilon}_{\boldsymbol{\xi}}(\boldsymbol{\theta}^{\mathrm{a}}_{n,t}, \boldsymbol{w}_n, t) = (1+\gamma)\epsilon_{\boldsymbol{\xi}}(\boldsymbol{\theta}^{\mathrm{a}}_{n,t}, \boldsymbol{w}_n, t) - \gamma\epsilon_{\boldsymbol{\xi}}(\boldsymbol{\theta}^{\mathrm{a}}_{n,t}, t),$$

$$\boldsymbol{\theta}^{\mathrm{a}}_{n,t-1} = \frac{1}{\sqrt{\alpha_t}}\left[\boldsymbol{\theta}^{\mathrm{a}}_{n,t} - \frac{\beta_t}{\sqrt{1-\overline{\alpha}_t}}\tilde{\epsilon}_{\boldsymbol{\xi}}(\boldsymbol{\theta}^{\mathrm{a}}_{n,t}, \boldsymbol{w}_n, t)\right] + \sigma_t \boldsymbol{z}_t, \quad (9)$$

where $\boldsymbol{z}_t \sim \mathcal{N}(\mathbf{0}, \boldsymbol{I})$ for $t > 1$ and $\boldsymbol{z}_t = \mathbf{0}$ for $t = 1$, $\beta_t = 1 - \alpha_t$, $\gamma \in [0,1]$. Alg. 2 in Appendix illustrates the detailed procedure.

Specifically, after generating the adapter parameter matrix, we reshape it to obtain the adapter parameters (for simplicity, we do not distinguish between the notations used before and after the reshaping). The generated adapter parameter is directly load into the adapter architecture. Then keeping the backbone parameters $\boldsymbol{\theta}^{\mathrm{b}}_n$ and the adapter parameters $\boldsymbol{\theta}^{\mathrm{a}}_{n,0}$ fixed, the recommendation model is directly applied to extract features from the user history interactions and score candidate items to generate a recommendation list that aligns with the preference weights of the new task.

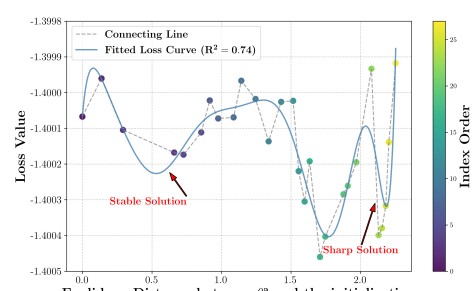

Figure 2: The relationship between the loss values in Eq. (7) and the adapter parameters $\boldsymbol{\theta}^{\mathrm{a}}_i$ (with the preference weights $w^1_i = 0.3$ and $w^2_i = 0.7$) on MovieLens 1M using SAS-Rec as backbone. The index order indicates the number of epochs after convergence. The detailed settings are shown in Sec. A.2

## 4 DISCUSSIONS ON THE ROBUSTNESS OF PARAMETER DIFFUSION

In experiments of Sec. 5, we found that parameter diffusion in our PaDiRec can generate model parameters that differ from those obtained by retraining the total loss in Eq. (7), yet still achieve good recommendation performance and controllability. This leads us to hypothesize that parameter diffusion may find more robust model parameters through model parameter generalization. To validate this hypothesis, we continued training the adapter parameters for multiple epochs after the total loss had converged, analyzing the relationship between the adapter parameters and the values of the loss Eq. (7). We employed polynomial fitting to model the relationship between the data points, estimating the functional relationship between the loss values and model parameters, achieving a goodness of fit of $R^2 = 0.74$. As illustrated in Figure 2, we observe that the solutions of the total loss exhibit different characteristics under various model parameters. Specifically, there are relatively flat solution sets, referred to as "Stable Solutions", as well as solution sets with more dramatic fluctuations, termed "Sharp Solutions". Furthermore, as verified in RQ1 of Sec. 5.3, we observe that the parameter diffusion in PaDiRec effectively learns the set of robust "Stable Solutions", thereby enhancing controllability while maintaining high performance.

Table 1: Performance comparison between the proposed method and baseline models. The **best** results are highlighted in bold, while the second-best results are underlined.

| Backbone | Algorithm | MovieLens Metrics | | | Amazon Food Metrics | | | Industrial Dataset Metrics | | |
|---|---|---|---|---|---|---|---|---|---|---|
| | | Avg.HV | Pearson r-a | Pearson r-d | Avg.HV | Pearson r-a | Pearson r-d | Avg.HV | Pearson r-a | Pearson r-d |
| SASRec | Retrain | **0.2281** | - | - | 0.2251 | - | - | 0.2779 | - | - |
| | CMR | 0.1920 | 0.8901 | 0.9150 | 0.1955 | -0.7039 | **0.9932** | 0.2476 | 0.8750 | 0.9237 |
| | Soup | 0.1441 | 0.7861 | 0.9133 | 0.1561 | 0.5317 | 0.6693 | 0.1825 | 0.7306 | 0.8188 |
| | MMR | 0.1808 | 0.9575 | 0.8803 | 0.1707 | 0.1320 | -0.3087 | 0.2034 | 0.9077 | 0.9655 |
| | **PaDiRec** (Ours) | 0.2138 | **0.9905** | **0.9903** | **0.2420** | **0.8857** | 0.9816 | **0.2812** | **0.9976** | **0.9986** |
| - | LLM | 0.0625 | -0.0600 | 0.0994 | 0.1017 | 0.7296 | 0.8558 | 0.0372 | 0.7279 | 0.7132 |
| GRU4Rec | Retrain | 0.1823 | - | - | 0.1556 | - | - | 0.1735 | - | - |
| | CMR | 0.1760 | 0.9068 | 0.8813 | **0.3617** | 0.8287 | 0.9059 | 0.1230 | 0.6514 | 0.5985 |
| | Soup | 0.1197 | 0.8061 | 0.6694 | 0.0604 | 0.3850 | 0.8005 | 0.1226 | 0.7099 | 0.8200 |
| | MMR | 0.1609 | 0.8692 | 0.7257 | 0.1354 | -0.3497 | -0.3748 | 0.1287 | 0.7916 | 0.7553 |
| | **PaDiRec** (Ours) | **0.2009** | **0.9929** | **0.9786** | 0.1623 | **0.8470** | **0.9685** | **0.1871** | **0.9760** | **0.9236** |
| - | LLM | 0.0625 | -0.0716 | 0.0119 | 0.0667 | 0.7484 | 0.7904 | 0.0372 | 0.8088 | 0.7722 |
| TiSASRec | Retrain | 0.2301 | - | - | 0.2232 | - | - | 0.2777 | - | - |
| | CMR | 0.1769 | 0.9286 | 0.9903 | 0.2064 | -0.6279 | 0.9828 | **0.3315** | 0.8855 | 0.9300 |
| | Soup | 0.1483 | 0.8033 | 0.8858 | 0.1533 | 0.5342 | 0.6525 | 0.1811 | 0.7310 | 0.8248 |
| | MMR | 0.1815 | 0.8946 | 0.8684 | 0.1672 | 0.3057 | 0.2037 | 0.2060 | 0.9004 | 0.9565 |
| | **PaDiRec** (Ours) | **0.2532** | **0.9923** | **0.9914** | **0.2394** | **0.8759** | **0.9851** | 0.2862 | **0.9968** | **0.9984** |
| - | LLM | 0.0625 | -0.0663 | 0.0999 | 0.0667 | 0.7213 | 0.8499 | 0.0372 | 0.7373 | 0.7451 |

## 5 EXPERIMENTS

We conducted experiments to evaluate the performance of PaDiRec on two public datasets and an industrial dataset for sequential recommendation.

### 5.1 EXPERIMENT SETTINGS

**Dataset.** We used two public datasets, `MovieLens 1M` [1] and `Amazon Food` [2], and the `Industrial Data` from a electronics commercial store. Detailed descriptions of the datasets and preprocessing methods can be found in Appendix A.3.

**Baselines.** The baselines are as follows. Retraining as Eq. (2), which is considered optimal. Soup (Wortsman et al., 2022), a classic algorithm for model merging. MMR (Carbonell & Goldstein, 1998), a rule-based post-process policy. CMR (Chen et al., 2023), an re-rank algorighm utilizing hypernetwork to achieve dynamic preference of changing. LLM (Appendix A.11), a prompt-based method for controllable recommendation. Details about all the baselines are shown in Appendix A.4

**Metrics.** We propose evaluating performance from two dimensions. Specifically, we use Hyper-volume (HV) (Guerreiro et al., 2021) to measure the performance of the algorithm on each task, particularly in terms of the trade-offs between accuracy and diversity. The average HV (denoted as **Avg.HV**) across multiple tasks is used to assess the overall performance of the algorithm in balancing both objectives (accuracy and diversity). To eliminate the differences in scale between the two objectives, we normalize the performance on each objective. Additionally, we utilize the Pearson correlation coefficient to evaluate the alignment between the algorithm's performance across different tasks and the optimal model, providing insight into the algorithm's controllability. **Pearson r-a**, **Pearson r-d** measure the correlation between the algorithm and the optimal in terms of accuracy (NDCG@10) and diversity ($\alpha$-NDCG@10).

### 5.2 EXPERIMENTAL RESULTS

We conducted experiments to address the following two questions: i) How transferable is PaDiRec, specifically in terms of its ability to adapt to different backbone algorithms? ii) How does PaDiRec perform compared to other baselines on each specific task? The results are presented in Table 1.

To answer the first question, we used commonly adopted sequential recommendation models as backbones (e.g., SASRec (Kang & McAuley, 2018), GRU4Rec (Hidasi, 2015), and TiSASRec (Li et al., 2020b)) and conducted extensive experiments across three datasets. Specifically, we evaluated PaDiRec's performance under various task descriptions by measuring NDCG@10 and $\alpha$-NDCG@10. The accuracy weight $w_{acc.}$ varies from 0 to 1 in intervals of 0.1, with the corresponding

---

[1]https://grouplens.org/datasets/movielens/

[2]http://jmcauley.ucsd.edu/data/amazon/links.html

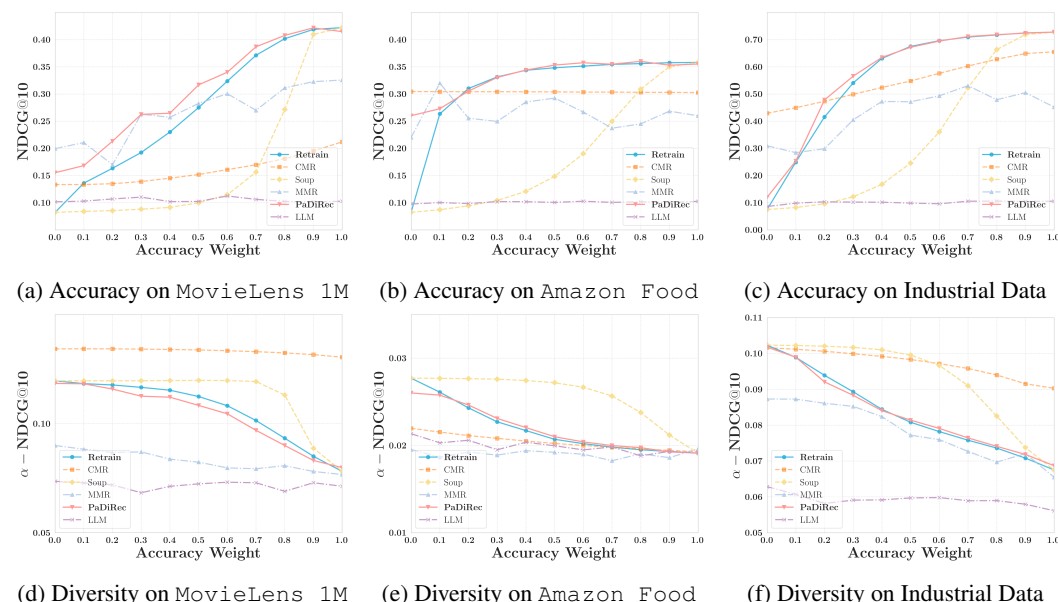

(a) Accuracy on `MovieLens 1M`  (b) Accuracy on `Amazon Food`  (c) Accuracy on Industrial Data

(d) Diversity on `MovieLens 1M`  (e) Diversity on `Amazon Food`  (f) Diversity on Industrial Data

Figure 3: The accuracy and diversity curve of PaDiRec and other baselines in NDCG@10 and $\alpha$-NDCG@10 across accuracy weights ranging from 0 to 1, with intervals of 0.1. The backbone is TiSASRec, results on other backbones are shown in Appendix A.6

diversity weight set as $w_{\text{div.}} = 1 - w_{\text{acc.}}$. We then post-processed NDCG@10 and $\alpha$-NDCG@10 across different tasks to compute Avg.HV, Pearson r-a, and Pearson r-d. These metrics respectively evaluate the quality of multi-objective optimization on individual tasks and the controllability across multiple tasks. Overall, across the three backbones and three datasets, PaDiRec consistently ranked among the top two performers across all three evaluation metrics. Notably, in most cases, the top two of Avg.HV are Retrain and PaDiRec, indicating that PaDiRec's performance in multi-objective trade-offs is on par with, or even superior to, Retrain. Specific exceptions occurred, such as on the `Amazon Food` dataset with GRU4Rec as the backbone and the `Industrial Data` with TiSASRec as the backbone, where CMR achieved the best Avg.HV. This is because CMR is not influenced by task descriptions and thus maintains consistently high NDCG@10 scores (as shown in Figure 3 and further explained in response to question ii). For Pearson r-a and Pearson r-d, PaDiRec demonstrated strong correlations with the Retrain method, indicating that PaDiRec closely aligns with Retrain (which we assume to be optima) in terms of accuracy (NDCG@10) and diversity ($\alpha$-NDCG@10) across different tasks. On the `Amazon Food` dataset with SASRec as the backbone, CMR achieved the highest Pearson r-d. However, its Pearson r-a was negative, indicating a lack of control and a collapse in accuracy.

To address the second question, we presented the specific performance of PaDiRec under each task description using TiSASRec as the backbone across three datasets, as shown in Figure 3 (more results are shown in Appendix. A.6). It is observed that in all three datasets, PaDiRec's NDCG@10 progressively increases with higher accuracy weights, while $\alpha$-NDCG@10 decreases correspondingly due to the simultaneous reduction in diversity weight. These trends demonstrate the effectiveness of our algorithm in controllability. Notably, assuming that Retrain is optimal, PaDiRec exhibits strong consistency with the Retrain method. In contrast, MMR, as a post-processing algorithm, shows variability because varying degrees of diversity manipulation can disrupt the original recommendation list, uncontrollably affecting its accuracy. The Soup method merges the parameters of accuracy and diversity models based on their weights, aligning closely with the Retrain model when accuracy weights are extreme but showing significant deviations in other tasks. This indicates that tasks do not follow a simple linear relationship with different preference weights, and Soup makes overly strong assumptions about this relationship. CMR demonstrates inconsistent performance across different datasets. On the `MovieLens 1M` dataset, CMR aligns well with the original descriptions by exhibiting high diversity. However, on the other two datasets, it shows a stable yet uncontrollable state; for instance, on the `Amazon Food` dataset, CMR maintains high accuracy

Table 2: Response time comparison between proposed PaDiRec and the "Retrain" approach across three datasets using three different backbones. Note that the unit is seconds (sec.).

| Approach | Backbone | MovieLens 1M (sec.) | Amazon Food (sec.) | Industrial Data (sec.) |
|---|---|---|---|---|
| | SASRec | 293.10 ± 11.61 | 91.01 ± 2.34 | 46.82 ± 3.25 |
| Retrain | GRU4Rec | 281.60 ± 17.36 | 92.39 ± 4.28 | 49.54 ± 2.38 |
| | TiSASRec | 303.80 ± 9.09 | 105.40 ± 7.66 | 52.47 ± 4.64 |
| | SASRec | 2.68 ± 0.36 | 2.64 ± 0.36 | 2.55 ± 0.25 |
| PaDiRec | GRU4Rec | 2.56 ± 0.27 | 2.54 ± 0.24 | 2.51 ± 0.23 |
| | TiSASRec | 2.55 ± 0.23 | 2.52 ± 0.24 | 2.58 ± 0.26 |

even with low accuracy weights, and on the `Industrial Data`, it retains high diversity despite low diversity weights.

## 5.3 ANALYSES

We conducted our analysis experiments based on three key research questions: **RQ1**: What are the advantages of Diffusion over Hypernetwork in parameter generation? **RQ2**: Is PaDiRec efficient enough to handle real-time changes in preference weights compared to Retrain? **RQ3**: How do different conditioning strategies impact the model's performance? Additionally, we also present some case study in Apprndix A.12.

**Regarding RQ1: Diffusion outperforms Hypernetwork in parameter generation.**

We conducted experiments to validate the robustness of the parameters generated by PaDiRec (as assumed in Sec. 4). Specifically, we designed three sets of experiments to constructed adapter parameters: "Retrain", "PaDiRec" and "Hypernetwork", where "Hypernetwork" utilizes the MLP to learn the relationship between the preference weight and the optimized adapter parameters.

First, we added Gaussian noise of the same magnitude to all three sets of adapter parameters and measured the resulting fluctuations in NDCG@10 and $\alpha$-NDCG@10. The experiments were repeated multiple times under various preference weights, and the results are shown in Figure 4. We observed that the parameters generated by PaDiRec exhibited the lowest performance fluctuations, both in terms of accuracy (NDCG@10) and diversity ($\alpha$-NDCG@10), when subjected to perturbations, which verifies the assumption in Sec. 4. Additionally, we compared the similarity (Inverse Euclidean Distance) between the three sets of parameters and the Retrain parameters. Hypernetwork-generated parameters showed slightly higher similarity to the Retrain parameters than those generated by Diffusion, with values of $0.3147 \pm 0.0184$

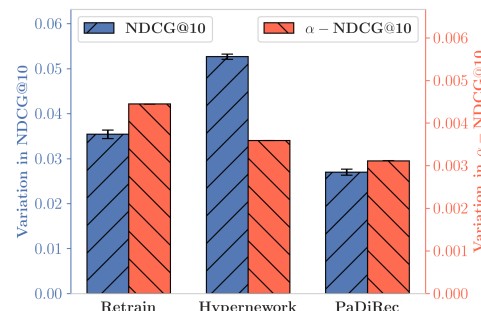

Figure 4: The variation of performance before and after disturbance in `MovieLens 1M` based on SASRec. The blue bars represent the variation in NDCG@10, while the red represent the variation in $\alpha$-NDCG@10

and $0.3035 \pm 0.0184$, respectively. This indicates that PaDiRec, as a diffusion-based parameter generator, is not merely mimicking parameters but has learned the underlying distribution of parameters, demonstrating its ability to generate robust, high-performance parameters (**?**).

**Regarding RQ2: The efficiency and effectiveness of PaDiRec.**

PaDiRec is designed to adaptively adjust model parameters in an online environment without retraining, enabling it to quickly respond to new task requirements. This places a strong emphasis on the model's response time. We compared the response times of PaDiRec and "Retrain" across various backbones and datasets, with the results shown in Table 2. As observed, in all experiments using three different backbones across three datasets, PaDiRec's response time was significantly faster than that of "Retrain". Notably, the response time of "Retrain" correlated with the size of the dataset, whereas PaDiRec exhibited minimal variation across different datasets. This highlights PaDiRec's data-agnostic nature indicating its potential for handling large-scale datasets efficiently.

**Regarding RQ3: Influence of different conditioning strategies.**

We investigate the influence of different conditioning strategies aimed at improving the integration of conditions into the denoising model. As shown in Figure 5, each strategy emphasizes different performance dimensions (details on the construction of each strategy can be found in Appendix A.5). In terms of Hypervolume, all five strategies outperform the "Retrain" approach, with the "Pre&Post" strategy achieving the best results. For Pearson r-a and Pearson r-d, the "Adap-norm" strategy demonstrates the best overall performance, indicating strong consistency with the "Retrain" approach, i.e., high controllability. Additionally, the Hypervolume remains within an acceptable range, suggesting that adding conditions aggregated by an attention mechanism to the layer norm is a promising approach for controllability.

## 6 RELATED WORK

**Diffusion models.** Diffusion probabilistic models (Ho et al., 2020; Song et al., 2020; Nichol & Dhariwal, 2021)have not only achieved significant success in the field of image generation but have also found wide applications in various other areas in recent years, such as video generation (Ho et al., 2022b), text generation (Li et al., 2022; Gong et al., 2022), etc. Moreover, diffusion models have shown the ability to generate high-quality neural network parameters, achieving comparable or even superior performance to traditionally trained models (Yuan et al., 2024; **?**). These models have also been applied to enhance the accuracy of recommender systems by addressing challenges such as noisy interactions and temporal shifts in user preferences (Wang et al., 2023). In our work, we utilize diffusion models to generate parameters for controllable multi-task recommender systems.

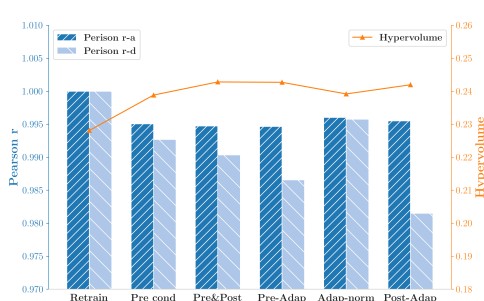

Figure 5: Performances of different conditioning strategies on `MovieLens 1M` using SASRec as backbone. The results of the "Retrain" algorithm are used as a reference.

**Multi-task learning (MTL)** aims to develop unified models that tackle multiple learning tasks simultaneously while facilitating information sharing (Zhang & Yang, 2021; Ruder, 2017). Recent advancements in MTL include deep networks with various parameter sharing mechanisms (Misra et al., 2016; Long et al., 2017; Yang & Hospedales, 2016) and approaches treating MTL as a multi-objective optimization problem (Lin et al., 2019; Mahapatra & Rajan, 2020; Xie et al., 2021). These latter methods focus on identifying Pareto-efficient solutions across tasks, with significant applications in recommender systems (Jannach, 2022; Li et al., 2020a; Zheng & Wang, 2022) Researchers have explored different strategies, from alternating optimization of joint loss and individual task weights to framing the process as a reinforcement learning problem (Xie et al., 2021). The emphasis has shifted from optimizing specific preference weights to finding weights that achieve Pareto efficiency across objectives (Sener & Koltun, 2018; Lin et al., 2019). Some methods utilize attention mechanisms to dynamically allocate computational resources among tasks (Liu et al., 2019). More recent approaches, such as the CMR (**?**), utilize hypernetworks to learn the entire trade-off curve for MTL problems. However, our novel approach diverges from these existing methods by employing diffusion models to control model parameters at test time, potentially offering greater flexibility and adaptability in handling multi-task learning problems.

## 7 CONCLUSIONS

This paper addresses the critical challenge of adapting recommendation models to dynamic task requirements in real-world applications, where frequent retraining is impractical due to high computational costs. To tackle this problem, we propose **PaDiRec**, a novel controllable learning approach that enables efficient adaptation of model parameters without retraining by utilizing a diffusion model as a parameter generator. Our approach is model-agnostic, allowing it to integrate with existing recommendation models and enhance their controllability. PaDiRec provides a practical solution for real-time, customizable recommendations in achieving efficient, test-time adaptation.

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

# A APPENDIX

## A.1 AGORITHMS

---

**Algorithm 1** Parameter Diffusion Model Training

---

1: **Input**: Dataset $\{(\boldsymbol{\theta}_{a,0}^i, \boldsymbol{w}_i)\}_{i=1}^N$ denoted as $\mathcal{D}$      ▷ Dataset Preparation
2: **Initialize**: Learnable parameters $\boldsymbol{\xi}$ for $g$
3: **Repeat**:
4:     $(\boldsymbol{\theta}_{a,0}^i, \boldsymbol{w}_i) \sim \mathcal{D}$      ▷ Sample data with conditioning from the dataset
5:     $\boldsymbol{w}_i \leftarrow \varnothing$ with probability $p_{\text{uncond}}$ ▷ Randomly discard conditioning to train unconditionally
6:     $t \sim \text{Uniform}(1, \ldots, T)$      ▷ Sample diffusion step
7:     $\epsilon_t \sim \mathcal{N}(0, I)$      ▷ Sample a Gussian noise
8:     $\nabla_{\boldsymbol{\xi}} \| \epsilon - \epsilon_{\boldsymbol{\xi}} (\sqrt{\overline{\alpha}_t} \boldsymbol{\theta}_{a,0}^i + \sqrt{1 - \overline{\alpha}_t} \epsilon_t, \boldsymbol{w}_i, t) \|^2$      ▷ Optimization of denoising model
9: **Until converged**

---

**Algorithm 2** Test-time Parameter Generation

---

1: **Input**: preference weights of new task $n$, denoted as $\boldsymbol{w}_n$, Gaussian noise $\theta_{a,T}^n \sim \mathcal{N}(0, I)$
2: **Initialize**: Trained parameters $\boldsymbol{\xi}$ for $g$, guidance strength $\gamma$
3: **for** $t \in \{1, 2, \ldots, T\}$ **do**
4:     **if** $t > 1$ **then**
5:       $z_t \sim \mathcal{N}(0, I)$
6:     **else**
7:       $z_t = 0$
8:     **end if**
9:     $\tilde{\epsilon}_{\boldsymbol{\xi}}(\boldsymbol{\theta}_{a,t}^n, \boldsymbol{w}_n, t) = (1 + \gamma)\epsilon_{\boldsymbol{\xi}}(\boldsymbol{\theta}_{a,t}^n, \boldsymbol{w}_n, t) - \gamma \epsilon_{\boldsymbol{\xi}}(\boldsymbol{\theta}_{a,t}^n, t)$
10:     $\theta_{a,t-1}^n = \frac{1}{\sqrt{\alpha_t}}(\theta_{a,t}^n - \frac{\beta_t}{\sqrt{1-\overline{\alpha}_t}} \tilde{\epsilon}_{\boldsymbol{\xi}}(\boldsymbol{\theta}_{a,t}^n, \boldsymbol{w}_n, t)) + \sqrt{\beta_t} z_t$
11: **end for**

---

## A.2 SETTINGS OF EXPERIMENT IN DISCUSSION

In this experiment, we conducted tests on MovieLens 1M with preference weights set to $w_i^1 = 0.3$ and $w_i^2 = 0.7$, using SASRec as the backbone. The experiment followed the loss function in Eq. 4. We first trained the adapter to convergence (approximately 30 epochs), then continued training for several more epochs, recording the loss values and adapter parameters after each epoch. Each point in the figure represents a recorded value.

## A.3 DATASETS

### A.3.1 DATASET INTRODUCTION

**MovieLens-1M** [3] contains 1,000,209 anonymous ratings of approximately 3,900 movies, provided by 6,040 users who joined MovieLens in 2000. We sorted each user's browsing history chronologically and filtered out users with fewer than five interactions. As a result, 994,338 interactions, 6,034 users, and 3,125 items were used in the final dataset. Each interaction is formatted to include user ID, item ID, and timestamp. To evaluate the diversity of the recommendation list, we extracted the genre information for each movie from the meta-information. Each movie may belong to one or more of the 18 available genres.

**Amazon Grocery and Gourmet Food** [4] contains 151,254 anonymous reviews of 8,713 products by 14,681 users, spanning from August 09, 2000 to July 23, 2014. Since the items belong to 156 categories, with each item assigned to only one category, we used the GloVe (Pennington et al., 2014) to generate embeddings for each category. We then applied K-means clustering to group them

---

[3] https://grouplens.org/datasets/movielens/
[4] http://jmcauley.ucsd.edu/data/amazon/links.html

into 30 broader categories, allowing each item to belong to one or more of these 30 broader genres. The interaction format is the same as MovieLens-1M.

**The industrial dataset** is the user click dataset from a electronics commercial store in , spanning from July 24, 2024, to August 24, 2024. We processed the raw data, filtering out users with fewer than 20 interactions, and randomly selected the interaction histories of 1,000 users. Each interaction was formatted to match the structure used in MovieLens-1M. Notably, the filtered interactions covered several categories. We manually merged similar categories into 27 broader ones, allowing each item to belong to multiple categories, thereby supporting diversity in the dataset.

### A.3.2 DATASET SETTINGS

For the recommendation model setup, we used the ReChorus framework[5] for standardized processing. Regarding data partitioning strategy, we employed the Leave-One-Out approach. Specifically, for each user's interaction history, interactions were sorted by timestamp, with the last interaction designated as the test set, the second-to-last interaction as the validation set, and the remaining sequence as the training set. During the training stage, negative sampling was set to 9 items per positive interaction, while during testing, the full item set was used.

### A.4 BASELINES

PadiRec was compared with several algorithms that were constructed in the controllable multi-task recommendation scenarios, including: **Retraining** is performed using Linear Scalarization (Birge & Louveaux, 2011) based on each task description, with the assumption that the resulting model parameters represent the optimal solution. **Soup** (Wortsman et al., 2022) obtain a new model by averaging the parameters of fine-tuned models without requiring additional computation during inference. In our work, we fine-tuned two models on accuracy and diversity respectively, and then merged them linearly based on the task description. **MMR** (Carbonell & Goldstein, 1998) is a heuristic post-processing approach with the item selected sequentially according to maximal marginal relevance. We set the hyper-parameters based on the task description to achieve varying degrees of diversity in the recommendations. **CMR** (Chen et al., 2023) dynamically adjusts models based on preference weights using policy hypernetworks to generate model parameters. **LLM** (the prompt is shown in A.11) is utilized as a personalized recommender system. We achieve controllable recommendations by inputting prompts containing specific preference weights to respond to users' real-time preferences. For our experiments, we selected the llama3-7B-Instruct model. Details regarding the prompts and settings can be found in the appendix.

### A.5 CONDITION STRATEGIES

**Pre Conditioning** (i.e., Pre cond.) "Pre" denotes that the preference weights embeddings are integrated into the parameters embeddings before being fed into self-attention layers. In this method, we simply add the preference weight embeddings to the parameters embeddings within the input sequence.

**Pre and Post Conditioning** (i.e., Pre&Post) "Post" denotes that the preference weights embeddings are interged after the parameters embeddings fed into self-attention layers. In this method, we add the preference weights embeddings both "Pre" and "Post".

---

[5]https://github.com/THUwangcy/ReChorus

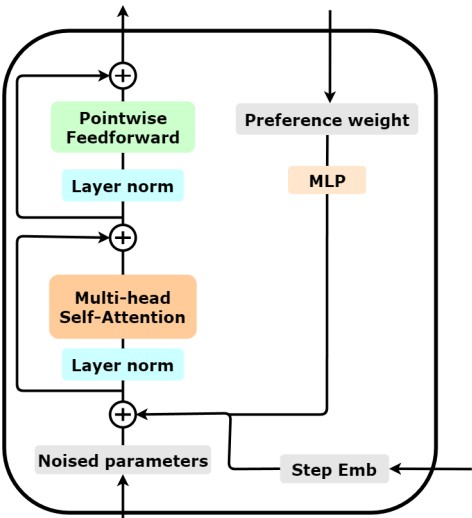

Figure 6: Illustration of conditioning strategies : Pre Conditioning

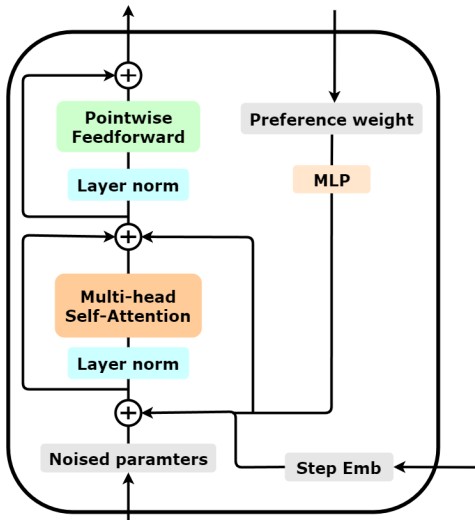

Figure 7: Illustration of conditioning strategies : Pre and Post Conditioning

**Pre-Adaptive Conditioning** (i.e., Pre-Adap) In this variant, we introduce an attention mechanism, which determines to what extent the preference weighted embeddings should be added to specific parameters embeddings. This approach aims to empower the model to learn how to adaptively utilize the preference weighted, enhancing its conditioning capabilities.

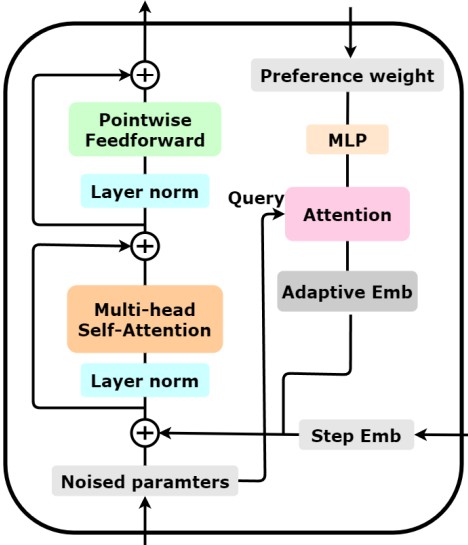

Figure 8: Illustration of conditioning strategies : Pre-Adaptive Conditioning

**Post-Adaptive Conditioning** (i.e., Post-Ada) The preference weights embeddings based on the attention mechanism is added after the multi-head self-attention in each transformer layer. Specifically, the query used for preference weights attentive aggregation is the output of the multi-head self-attention layer.

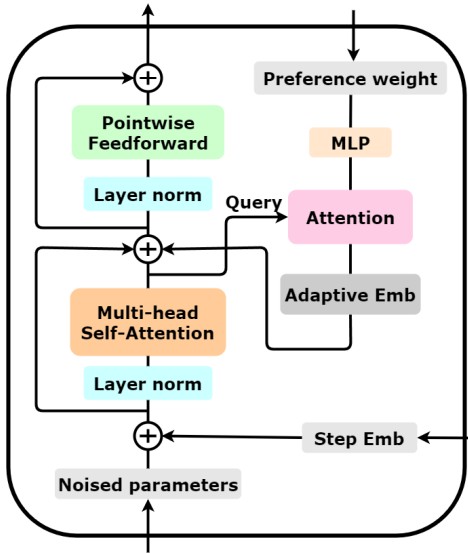

Figure 9: Illustration of conditioning strategies : Post-Adaptive Conditioning

**Adaptive-Norm Conditioning** (i.e., Ada-Norm) The preference weights embeddings based on the attention mechanism is used for re-scaling the output in each layer norm.

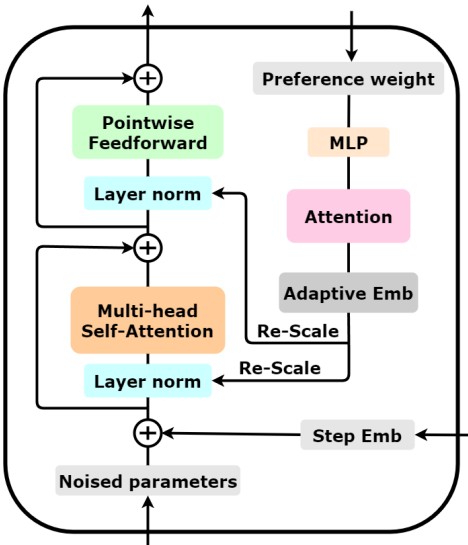

Figure 10: Illustration of conditioning strategies : Adaptive-Norm Conditioning

### A.6 THE CURVE OF ACCURACY (NDCG@10) AND DIVERSITY ($\alpha$-NDCG@10) ON OTHER BACKBONES

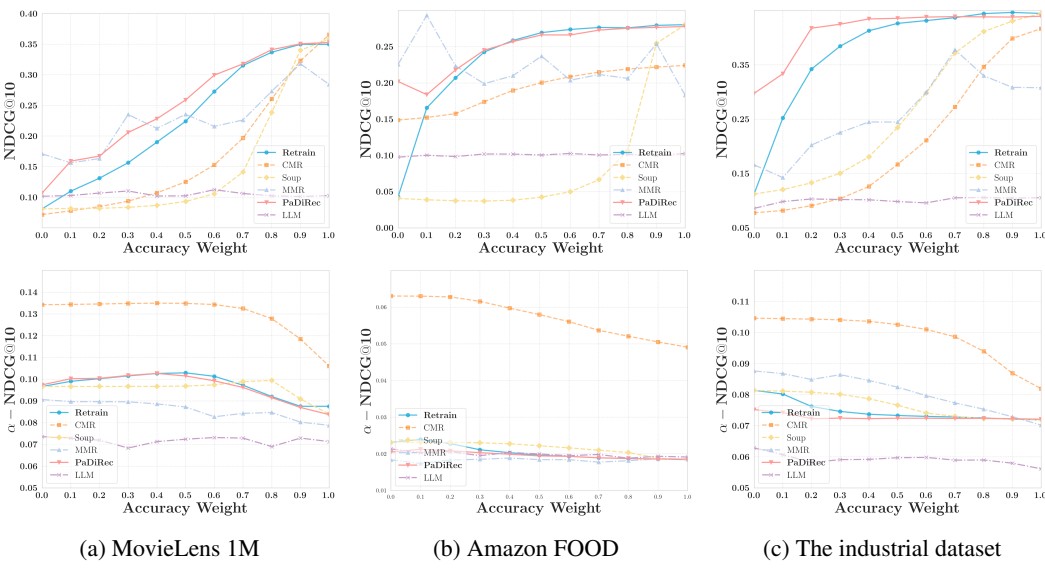

Figure 11: The trend of PadiRec and other baselines in NDCG@10 and $\alpha$-NDCG@10 across accuracy weights ranging from 0 to 1, with intervals of 0.1. The backbone is GRU4Rec

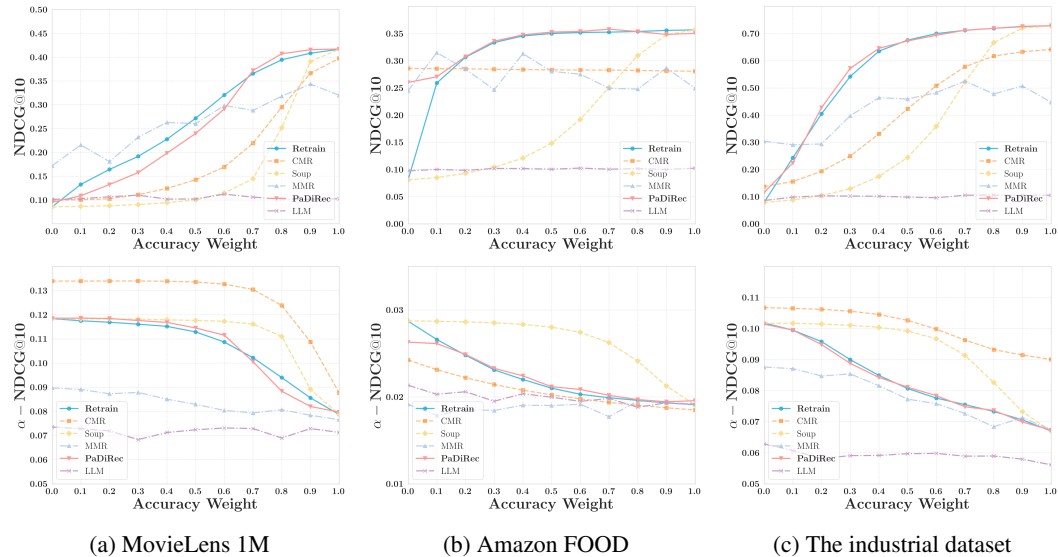

(a) MovieLens 1M       (b) Amazon FOOD       (c) The industrial dataset

Figure 12: The trend of PadiRec and other baselines in NDCG@10 and $\alpha$-NDCG@10 across accuracy weights ranging from 0 to 1, with intervals of 0.1. The backbone is SASRec

## A.7 SOTA BACKBONE (LRUREC)

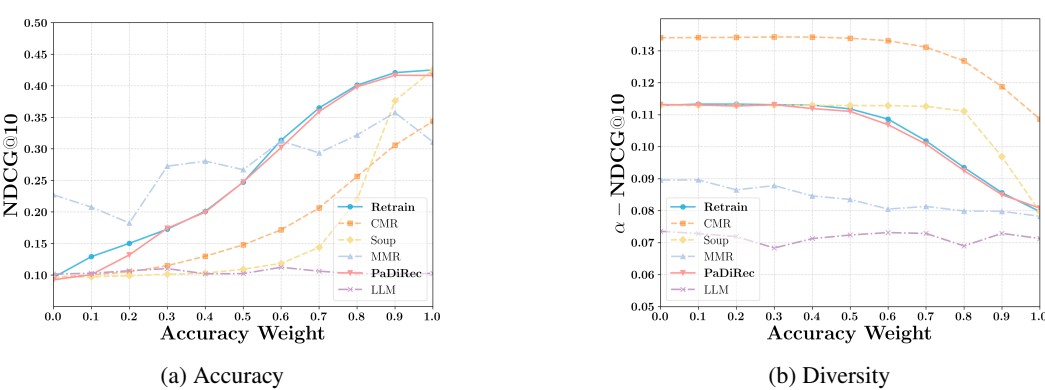

(a) Accuracy            (b) Diversity

Figure 13: The trend of PadiRec and other baselines in NDCG@10 and $\alpha$-NDCG@10 across accuracy weights ranging from 0 to 1, with intervals of 0.1. The backbone is **LRURec**

## A.8 THE EMBEDDING SIZE PROBLEM

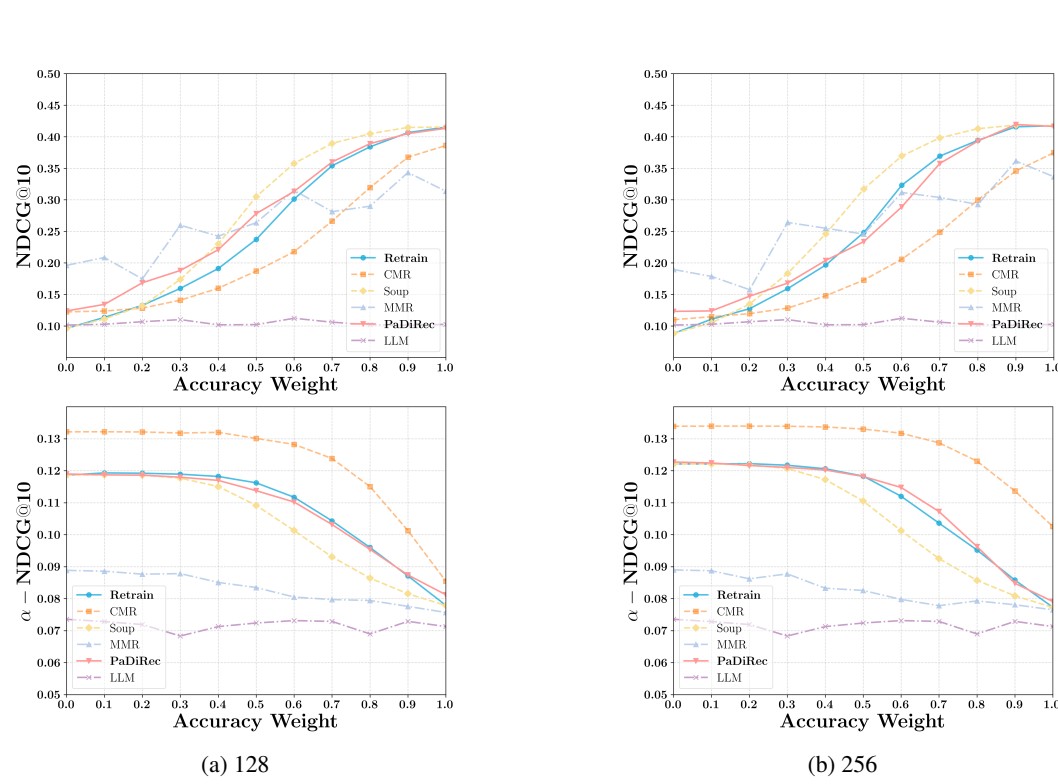

(a) 128

(b) 256

Figure 14: The trend of PadiRec and other baselines in NDCG@10 and $\alpha$-NDCG@10 across accuracy weights ranging from 0 to 1, with intervals of 0.1. The backbone is SASRec.

## A.9 MORE OBJECTIVES (ACCURACY, DIVERSITY, FAIRNESS)

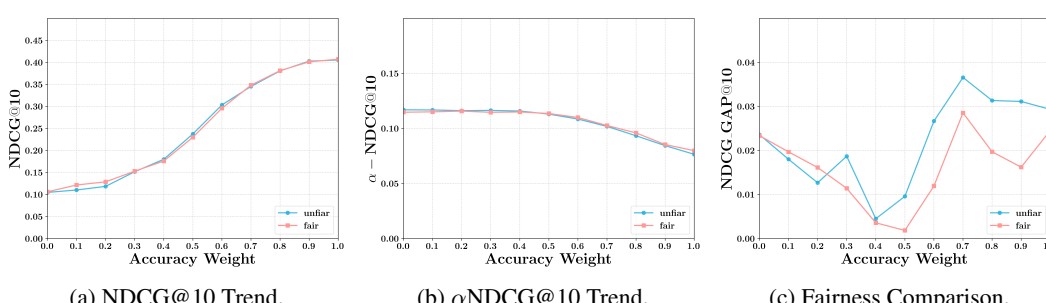

(a) NDCG@10 Trend.   (b) $\alpha$NDCG@10 Trend.   (c) Fairness Comparison.

Figure 15: The comparison of PadiRec between 'fair' and 'unfair' in metrics NDCG_GAP@10(fairness), NDCG@10(accuracy) and $\alpha$-NDCG@10(diversity) across accuracy weights ranging from 0 to 1, with intervals of 0.1. The backbone is SASRec. The dataset is Movielens. Note that a smaller NDCG_GAP@10 indicates a smaller difference in NDCG@10 between male and female user groups, signifying greater fairness.

Table 3: Fine-grained comparison of NDCG@10(accuracy), $\alpha$-NDCG@10(diversity), and NDCG_GAP@10(fairness) under different fairness weights while keeping the accuracy weight and diversity weight fixed.

| Acc. weight | Metric | Fair. weight = 0.1 | 0.4 | 0.7 | 1.0 |
|---|---|---|---|---|---|
| 0.6 | NDCG@10 | 0.3034 | 0.2910 | 0.2945 | 0.2959 |
| | a-NDCG@10 | 0.1085 | 0.1096 | 0.1094 | 0.1100 |
| | NDCG_GAP@10 | 0.0267 | 0.0253 | 0.0175 | 0.0119 |
| 0.7 | NDCG@10 | 0.3455 | 0.3448 | 0.3395 | 0.3482 |
| | a-NDCG@10 | 0.1019 | 0.1019 | 0.1045 | 0.1027 |
| | NDCG_GAP@10 | 0.0366 | 0.0299 | 0.0286 | 0.0285 |

## A.10 Network Layers of Recommendation Models

In our experiments, we implement our framework on three recommendation models, SASRec (Kang & McAuley, 2018), GRU4Rec (Hidasi, 2015), and TiSASRec (Li et al., 2020b). We provide their details of parameter structure in Tables 4, 5, and 6 respectively.

**SASRec (Kang & McAuley, 2018)** Self-Attentive Sequential Recommendation. This model employs a Transformer architecture to model user sequences for personalized recommendation tasks. It utilizes self-attention mechanisms that capture both long and short-term preferences by attending differently to items based on their relevance.

**GRU4Rec (Hidasi, 2015)** Gated Recurrent Units for Recommendation Systems. GRU4Rec leverages gated recurrent units (GRUs) to model user interaction sequences for session-based recommendations. By utilizing a gating mechanism, it effectively captures dependencies across varying time gaps between interactions, making it robust to session shifts and dropout behaviors.

**TiSASRec (Li et al., 2020b)** Time Interval-Aware Self-Attention for Sequential Recommendation. This model extends SASRec by incorporating time intervals between user interactions as an additional context. TiSASRec modifies the self-attention mechanism to account for these intervals, providing a more nuanced understanding of user preferences that evolve over time. The model includes a specialized positional encoding scheme to integrate these time dynamics alongside the sequential user behaviors.

**LRURec (Yue et al., 2024)** Linear Recurrent Units for Sequential Recommendation. This model introduces a novel linear recurrent unit architecture tailored for sequential recommendation tasks. LRURec combines the efficiency of recurrent neural networks with the modeling capabilities of self-attention mechanisms, enabling rapid inference and incremental updates on sequential data. By decomposing linear recurrence operations and implementing recursive parallelization, LRURec achieves reduced model size and parallelizable training.

Table 4: Parameter structure of SASRec(n depends on dataset)

| Layer name | Parameter shape | Parameter count |
|---|---|---|
| i_embeddings.weight | [n, 64] | 64n |
| p_embeddings.weight | [21, 64] | 1344 |
| transformer_block.0.masked_attn_head.q_linear.weight | [64, 64] | 4096 |
| transformer_block.0.masked_attn_head.q_linear.bias | [64] | 64 |
| transformer_block.0.masked_attn_head.k_linear.weight | [64, 64] | 4096 |
| transformer_block.0.masked_attn_head.k_linear.bias | [64] | 64 |
| transformer_block.0.masked_attn_head.v_linear.weight | [64, 64] | 4096 |
| transformer_block.0.masked_attn_head.v_linear.bias | [64] | 64 |
| transformer_block.0.layer_norm1.weight | [64] | 64 |
| transformer_block.0.layer_norm1.bias | [64] | 64 |
| transformer_block.0.linear1.weight | [64, 64] | 4096 |
| transformer_block.0.linear1.bias | [64] | 64 |
| transformer_block.0.linear2.weight | [64, 64] | 4096 |
| transformer_block.0.linear2.bias | [64] | 64 |
| transformer_block.0.layer_norm2.weight | [64] | 64 |
| transformer_block.0.layer_norm2.bias | [64] | 64 |
| adapter.0.weight | [8, 64] | 512 |
| adapter.0.bias | [8] | 8 |
| adapter.2.weight | [64, 8] | 512 |
| adapter.2.bias | [64] | 64 |

Table 5: Parameter structure of GRU4Rec(n depends on dataset)

| Layer name | Parameter shape | Parameter count |
|---|---|---|
| i_embeddings.weight | [n, 64] | 64n |
| rnn.weight_ih_l0 | [192, 64] | 12288 |
| rnn.weight_hh_l0 | [192, 64] | 12288 |
| rnn.bias_ih_l0 | [192] | 192 |
| rnn.bias_hh_l0 | [192] | 192 |
| out.weight | [64, 64] | 4096 |
| out.bias | [64] | 64 |
| adapter.0.weight | [8, 64] | 512 |
| adapter.0.bias | [8] | 8 |
| adapter.2.weight | [64, 8] | 512 |
| adapter.2.bias | [64] | 64 |

Table 6: Parameter structure of TiSASRec(n depends on dataset)

| Layer name | Parameter shape | Parameter count |
|---|---|---|
| i_embeddings.weight | [n, 64] | 64n |
| p_k_embeddings.weight | [21, 64] | 1344 |
| p_v_embeddings.weight | [21, 64] | 1344 |
| t_k_embeddings.weight | [513, 64] | 32832 |
| t_v_embeddings.weight | [513, 64] | 32832 |
| transformer_block.0.masked_attn_head.v_linear.weight | [64, 64] | 4096 |
| transformer_block.0.masked_attn_head.v_linear.bias | [64] | 64 |
| transformer_block.0.masked_attn_head.k_linear.weight | [64, 64] | 4096 |
| transformer_block.0.masked_attn_head.k_linear.bias | [64] | 64 |
| transformer_block.0.masked_attn_head.q_linear.weight | [64, 64] | 4096 |
| transformer_block.0.masked_attn_head.q_linear.bias | [64] | 64 |
| transformer_block.0.layer_norm1.weight | [64] | 64 |
| transformer_block.0.layer_norm1.bias | [64] | 64 |
| transformer_block.0.linear1.weight | [64, 64] | 4096 |
| transformer_block.0.linear1.bias | [64] | 64 |
| transformer_block.0.linear2.weight | [64, 64] | 4096 |
| transformer_block.0.linear2.bias | [64] | 64 |
| transformer_block.0.layer_norm2.weight | [64] | 64 |
| transformer_block.0.layer_norm2.bias | [64] | 64 |
| adapter.0.weight | [8, 64] | 512 |
| adapter.0.bias | [8] | 8 |
| adapter.2.weight | [64, 8] | 512 |
| adapter.2.bias | [64] | 64 |

## A.11 Prompt of LLM in Controllable Multi-Task Recommendation

"You are a recommendation expert who receives a user's chronological purchase history and provides the next recommended item from a given set of candidate products. " "Please note that you need to consider the äccuracyänd d̈iversityöf recommendations comprehensively.. " "Their definitions are as follows: " "Objective 1: Accuracy: Ensure that the recommended items are highly relevant to the user's interests and needs, thereby ensuring the accuracy of the recommendations. To measure the effectiveness of your recommendations, it will use nDCG as the evaluation metric. " "Objective 2: Diversity: Ensure that the recommended content is diverse, avoiding excessive recommendations of similar items. To achieve this, it will use Alpha-nDCG as the evaluation metric, penalizing overly similar recommended items and encouraging a diverse range of content in the recommendation list. " "Now, I will provide the current user's purchase history and the set of candidate products. Purchase history: [-history-]. Candidate product set: [-candidates-]." "Please rank these candidates and give [-out_num-] item as recommendationns and make them both diverse and accurately relevant with the history preference. " "To achieve this, consideri the priority of these two objectives according to the given priority weights (äccuracy:̈[-w_accuracy-] and d̈iversity:̈[-w_diversity-]) " "Split your output with line break. You MUST rank and output 10 items as recommendations. " "You can not generate candidates that are not in the given candidate set."

## A.12 CASE STUDY

In this section, we present the specific recommendation performance in each set of preference weights. We randomly select one customer from each of the three datasets and demonstrate the performance of the recommendation system models on these individual customers. As the accuracy weight increases (i.e., the diversity weight decreases), we observe a downward trend in the height of the bar chart, indicating that the number of categories represented in the recommendation list decreases, signifying a reduction in diversity. Meanwhile, the line chart shows an upward trend, suggesting that the target item's rank moves higher, reflecting an improvement in recommendation accuracy. The specific details are illustrated in the figures below. These changes clearly demonstrate the effectiveness of the PadiRec algorithm in controllable multi-task recommendation.

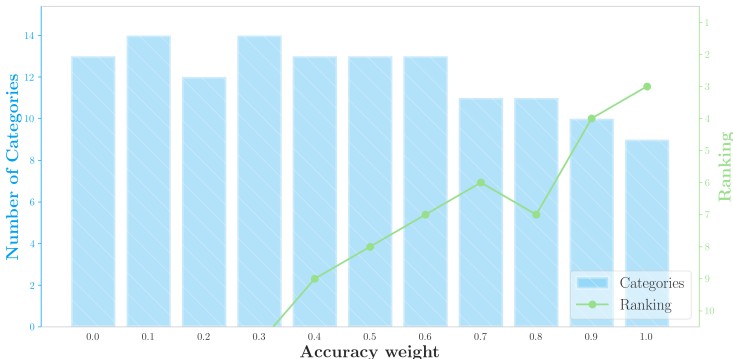

Figure 16: Case study on MovieLens 1M utilizing SASRec as the backbone. Note that the bars represent all the categories contained in the top-10 item list (some lists may contain more than 10 categories, as one item can belong to multiple categories). The line chart represents the rank of the target item in each list.

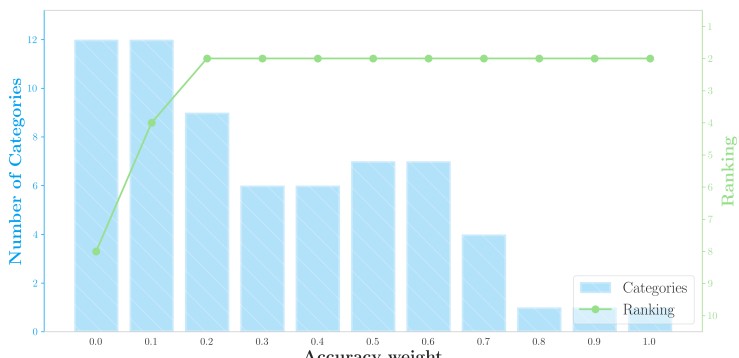

Figure 17: Case study on Amazon FOOD utilizing SASRec as the backbone. Note that the bars represent all the categories contained in the top-10 item list (some lists may contain more than 10 categories, as one item can belong to multiple categories). The line chart represents the rank of the target item in each list.

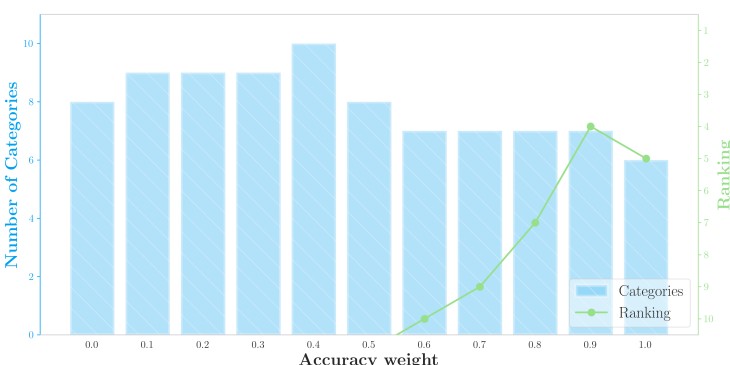

Figure 18: Case study on The industrial dataset utilizing SASRec as the backbone. Note that the bars represent all the categories contained in the top-10 item list (some lists may contain more than 10 categories, as one item can belong to multiple categories). The line chart represents the rank of the target item in each list.

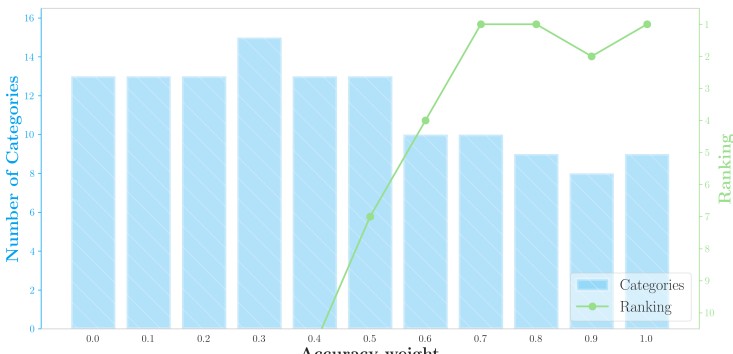

Figure 19: Case study on the MovieLens-1M dataset utilizing GRU4Rec as the backbone. Note that the bars represent all the categories contained in the top-10 item list (some lists may contain more than 10 categories, as one item can belong to multiple categories). The line chart represents the rank of the target item in each list.

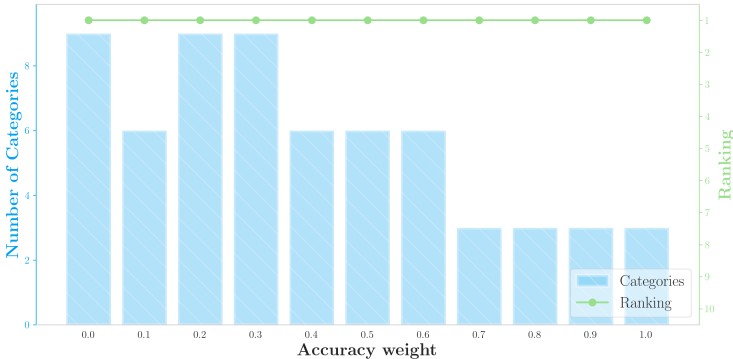

Figure 20: Case study on the Amazon FOOD dataset utilizing GRU4Rec as the backbone. Note that the bars represent all the categories contained in the top-10 item list (some lists may contain more than 10 categories, as one item can belong to multiple categories). The line chart represents the rank of the target item in each list.

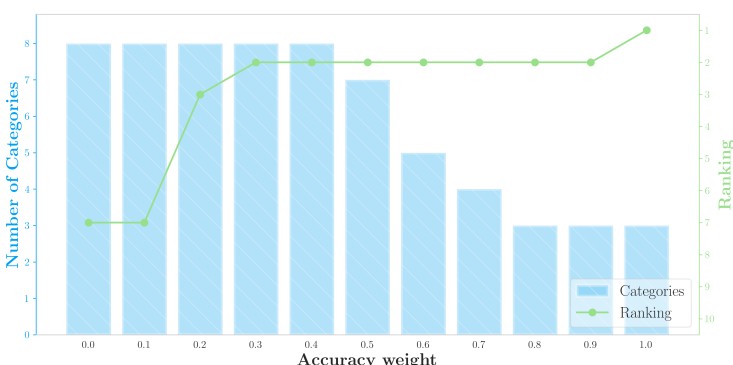

Figure 21: Case study on the industrial dataset utilizing GRU4Rec as the backbone. Note that the bars represent all the categories contained in the top-10 item list (some lists may contain more than 10 categories, as one item can belong to multiple categories). The line chart represents the rank of the target item in each list.

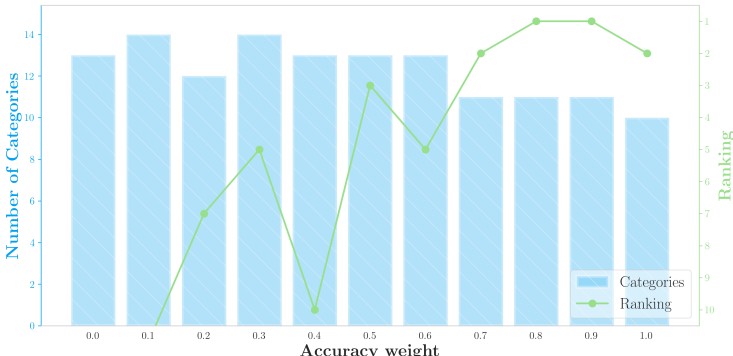

Figure 22: Case study on MovieLens 1M dataset utilizing TiSASRec as the backbone. Note that the bars represent all the categories contained in the top-10 item list (some lists may contain more than 10 categories, as one item can belong to multiple categories). The line chart represents the rank of the target item in each list.

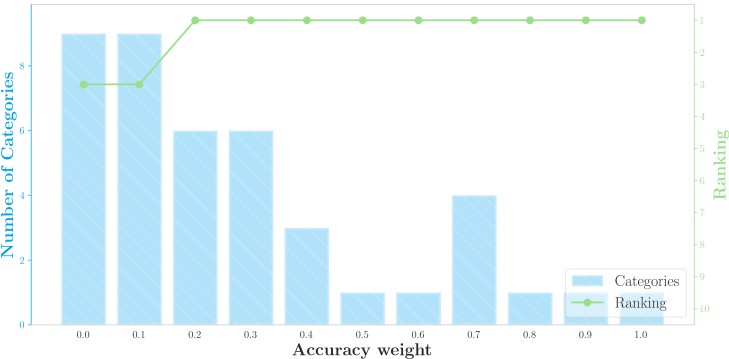

Figure 23: Case study on Amazon FOOD dataset utilizing TiSASRec as the backbone. Note that the bars represent all the categories contained in the top-10 item list (some lists may contain more than 10 categories, as one item can belong to multiple categories). The line chart represents the rank of the target item in each list.

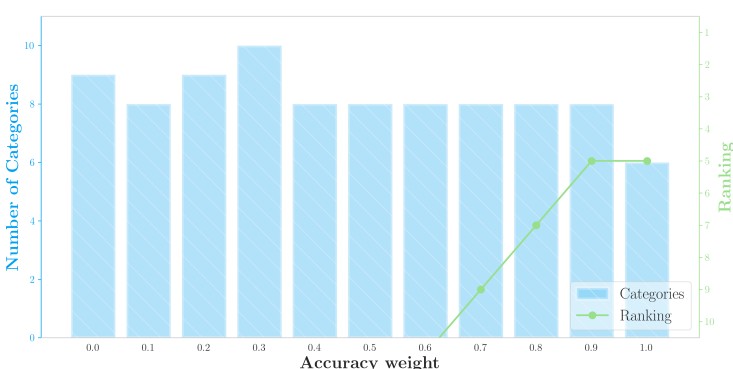

Figure 24: Case study on the industrial dataset utilizing TiSASRec as the backbone. Note that the bars represent all the categories contained in the top-10 item list (some lists may contain more than 10 categories, as one item can belong to multiple categories). The line chart represents the rank of the target item in each list.

Table 7: Case Study of PaDiRec on MovieLens 1M utilizing SASRec as the backbone. We compared the top-10 recommendation lists between an accuracy weight of 0.1 and an accuracy weight of 0.9. Notably, when the accuracy weight is 0.1 (indicating a high preference for diversity), items covering more categories are ranked higher, but the list does not include the target item, indicating poor accuracy. Conversely, with an accuracy weight of 0.9, the target item is ranked in the top 1 position within the recommendation list.

| Accuracy | Category | Item | Is Target Item |
|---|---|---|---|
| 0.1 | Animation, Children's, Comedy, Musical, Romance | Little Mermaid | No |
| 0.1 | Action, Comedy, Crime, Horror, Thriller | From Dusk Till Dawn | No |
| 0.1 | Adventure, Fantasy, Sci-Fi | Time Bandits | No |
| 0.1 | Animation, Children's | Sword in the Stone | No |
| 0.1 | Action, Romance, Thriller | Desperado | No |
| 0.1 | Adventure, Children's, Fantasy | Santa Claus | No |
| 0.1 | Horror, Sci-Fi | Invasion of the Body Snatchers | No |
| 0.1 | Film-Noir, Mystery, Thriller | Palmetto | No |
| 0.1 | Action, Comedy | Twin Dragons | No |
| 0.1 | Film-Noir | Sunset Blvd. | No |
| 0.9 | Horror | Birds | Yes |
| 0.9 | Drama | Cider House Rules | No |
| 0.9 | Comedy, Romance | Annie Hall | No |
| 0.9 | Action, Comedy, Crime, Horror, Thriller | From Dusk Till Dawn | No |
| 0.9 | Drama, Romance | Girl on the Bridge | No |
| 0.9 | Animation, Children's, Comedy, Musical, Romance | Little Mermaid | No |
| 0.9 | Comedy | Road Trip | No |
| 0.9 | Comedy, Drama | Chuck & Buck | No |
| 0.9 | Horror, Sci-Fi | Invasion of the Body Snatchers | No |
| 0.9 | Animation, Children's | Sword in the Stone | No |

## A.13 DETAILS OF DIFFUSION

Table 8: Parameter structure of Model, where the transformer block index 'x' ranges from 0 to 3.

| Layer name | Parameter shape | Parameter count |
|---|---|---|
| embedding.weight | [8, 512] | 4096 |
| linear.weight | [512, 8] | 4096 |
| linear.bias | [8] | 8 |
| transformer_block.x.self_attn_out_proj.weight | [512, 512] | 262144 |
| transformer_block.x.self_attn_out_proj.bias | [512] | 512 |
| transformer_block.x.linear1.weight | [512, 2048] | 1048576 |
| transformer_block.x.linear1.bias | [2048] | 2048 |
| transformer_block.x.linear2.weight | [2048, 512] | 1048576 |
| transformer_block.x.linear2.bias | [512] | 512 |
| transformer_block.x.norm1.weight | [512] | 512 |
| transformer_block.x.norm1.bias | [512] | 512 |
| transformer_block.x.norm2.weight | [512] | 512 |
| transformer_block.x.norm2.bias | [512] | 512 |
| transformer_block.x.dropout1.weight | [512] | 512 |
| transformer_block.x.dropout1.bias | [512] | 512 |
| transformer_block.x.dropout2.weight | [512] | 512 |
| transformer_block.x.dropout2.bias | [512] | 512 |
| step_mlp.0.weight | [512, 512] | 262144 |
| step_mlp.0.bias | [512] | 512 |
| step_mlp.1.weight | [512, 512] | 262144 |
| step_mlp.1.bias | [512] | 512 |
| step_mlp.2.weight | [512, 512] | 262144 |
| step_mlp.2.bias | [512] | 512 |
| kgEmb_mlp.0.weight | [2, 512] | 1024 |
| kgEmb_mlp.0.bias | [512] | 512 |
| timeEmb_mlp.0.weight | [2, 512] | 1024 |
| timeEmb_mlp.0.bias | [512] | 512 |

## A.14 DIFFUSION TRANSFORMER FLOPS CALCULATION.

### A.14.1 BASE PARAMETERS

- Sampling steps ($T$): 500
- Input shape: $[batch\_size, channels, sequence\_length] = [1, 8, 137]$
- Condition vector: $2 \times 1$
- Model dimension ($d\_model$): 512
- Number of transformer layers ($N$): 4
- Number of attention heads: 8

### A.14.2 SINGLE STEP COMPUTATION BREAKDOWN

1. Initial Processing

- Input permute: $[1, 8, 137] \rightarrow [1, 137, 8]$
- Linear projection ($8 \rightarrow 512$): $137 \times (8 \times 512) = 561,152$ FLOPs
- Condition embedding ($2 \rightarrow 512$): $2 \times 512 = 1,024$ FLOPs
- Time embedding ($2 \rightarrow 512$): $2 \times 512 = 1,024$ FLOPs
- Step embedding: $2 \times (512 \times 512) + 512 = 524,800$ FLOPs
- **Total Initial Processing:** $1,088,000$ FLOPs

2. Transformer Layer Computation (per layer)

- Input shape: $[137, 512]$
- Self-Attention computation:

$$\text{Query, Key, Value projections: } 3 \times (137 \times 512 \times 512) = 107,479,040 \text{ FLOPs}$$
$$\text{Attention score computation: } (137 \times 137 \times 64) \times 8 = 12,055,552 \text{ FLOPs}$$
$$\text{Value weighting: } (137 \times 137 \times 64) \times 8 = 12,055,552 \text{ FLOPs}$$
$$\text{Output projection: } 137 \times 512 \times 512 = 35,826,688 \text{ FLOPs}$$

- Feed-forward network computation:

$$\text{First linear layer } (512 \rightarrow 2048): 137 \times 512 \times 2048 = 143,065,088 \text{ FLOPs}$$
$$\text{Second linear layer } (2048 \rightarrow 512): 137 \times 2048 \times 512 = 143,065,088 \text{ FLOPs}$$
$$\text{GELU activation: } 137 \times 2048 = 280,576 \text{ FLOPs}$$

- **Single Transformer Layer Total:** $453,827,584$ FLOPs

3. Output Processing

- Final linear projection $(512 \rightarrow 8)$: $137 \times (512 \times 8) = 561,152$ FLOPs

### A.14.3 TOTAL COMPUTATION

1. Single Step Computation

- Initial processing: 1,088,000
- Transformer layers: $453,827,584 \times 4 = 1,815,310,336$
- Output processing: $561,152$
- **Per step total:** $1,816,959,488$ FLOPs

2. Complete Sampling Process (500 steps):

- **Total FLOPs:** $1,816,959,488 \times 500 = 908,479,744,000 \approx 0.9085$ TFLOPs

### A.14.4 EFFICIENCY EVALUATION AND CONCLUSION

Taking the RTX 3090 as an example, which achieves 35.58T FLOPS per second, our diffusion model requires only 0.9085T FLOPs for the entire 500-step sampling process. Therefore, the inference process of the diffusion model takes approximately 0.026 seconds. Including some data storage overhead, the total time is around the order of seconds (aligned with Table 2). In real-world recommendation scenarios, a single recommendation typically occurs within milliseconds. However, for users, waiting 2-3 seconds to customize a more personalized model is considered acceptable.

### A.15 DISCUSSION ON USED MOVIELENS EVALUATION

We utilized the MovieLens dataset for our experiments. MovieLens is among the most widely used datasets in the recommender systems domain, serving as a standard benchmark for validating new models and ensuring reproducibility. However, it is important to acknowledge that the user-item interactions recorded in the MovieLens dataset primarily reflect engagements between users and the MovieLens platform, where users are prompted to recall movies they have previously watched. This setup differs significantly from typical recommendation scenarios encountered in real-world applications, as analysis by Fan et al. (2024).

Nonetheless, the MovieLens dataset remains valuable for research purposes. It provides researchers with a standardized benchmark, facilitating the verification of model implementations and enabling comparisons with existing studies. Besides, it is important to note that, our focus is not on the

patterns of sequential data but rather on providing controllability to the backbone models. Therefore, employing the MovieLens dataset in our research is justified. We recognize the limitations of relying solely on the MovieLens dataset. To comprehensively assess the effectiveness and applicability of our model, we conducted experiments on three datasets, further demonstrating the effectiveness of the algorithm.

