# OpenReview forum: "Generating Model Parameters for Controlling: Parameter Diffusion for Controllable  Multi-Task Recommendation"
_ICLR.cc/2025/Conference — Submitted to ICLR 2025_

### Official Review · Reviewer_kFtK · 2024-10-27

**Soundness:** 2
**Presentation:** 2
**Contribution:** 2
**Rating:** 5
**Confidence:** 4

**Summary:**

This paper introduces PaDiRec, a neural network diffusion-based approach for generating adaptable model parameters for controllable multi-task recommendation.

**Strengths:**

1. This paper presents an intriguing application of diffusion models for multi-task recommendation.

2. The paper is well-written and organized.

3. Code availability enhances the paper’s reproducibility.

**Weaknesses:**

Here are the polished points based on the content of the paper:

1. This paper primarily represents an application of the paper of "Neural Network Diffusion"  [1] in recommender systems, specifically building on the framework from the referenced neural network diffusion work. While the authors cite this paper, they do not provide a thorough enough introduction to or comparison with this prior work in the related work section, which does not fully credit the original contribution. It is suggested that this work be submitted to an application-focused conference.

2. This paper emphasizes practical applications rather than algorithmic development, making online testing essential. However, all experiments are conducted on offline datasets, raising concerns about the real-world usability of the proposed approach.

3. The backbone models used, such as SASRec, are not state-of-the-art in recommendation systems, which could limit the effectiveness and generalizability of the proposed approach when compared with modern backbone models in recent two years such as [2].

4. The paper’s focus is limited to accuracy and diversity, yet recommendation systems are often evaluated by a broader set of metrics, including serendipity, fairness, and more. Expanding the testing metrics could more robustly validate the method's effectiveness across a range of evaluation criteria.

[1] Wang, Kai, et al. "Neural network diffusion." *arXiv preprint arXiv:2402.13144* (2024).

[2] Yue, Zhenrui, et al. "Linear recurrent units for sequential recommendation." *Proceedings of the 17th ACM International Conference on Web Search and Data Mining*. 2024.

**Questions:**

1. Are there references supporting the use of Pearson r-d, or is this application new in this paper? If it’s new, could other established methods be used for evaluation instead?

2. In calculating the reported efficiency, is the diffusion training time included into the results? Additionally, is the diffusion model designed to require only a single training session that allows for permanent deployment, or does it need periodic retraining?

---

> ### Author Response · Authors · 2024-11-19
>
> Thank you for your time and effort. Some of your suggestions are highly valuable and worth adopting. However, there are a few misunderstandings regarding the contributions of our paper and its relevance to the conference topic. We have provided detailed explanations and conducted additional experiments to address each of your concerns, and we hope this helps to clarify any confusion!
>
> # W1&W2: Contribution
> The work [1] provided us with significant inspiration, as it demonstrated the critical role of diffusion models in high-performance parameter reconstruction. However, in our work, the diffusion model serves as just one tool for parameter generation. It is necessary for us to restate our **key contributions** and innovations: (1) the **definition** of Controllable Multi-Task Recommendation (**CMTR**), (2) the "backbone + task-specific adapter" **structure** in recommendation models, and (3) the use of conditional training to achieve the **"one instruction, one model" paradigm**. We acknowledge that our paper does not include A/B testing results, due to the company's complex processes and confidentiality requirements. Nevertheless, we conducted tests using **real clickstream data** collected from July 24, 2024, to August 24, 2024, as presented in the paper (**Table 1**), which reflects a certain level of consistency with the online tests.
>
> Additionally, similar methodologies have already been adopted in transfer learning, as shown in work [2] published at ICLR 2024. This supports the relevance of adapting new technologies to different domains, **aligning with the conference topic**. Furthermore, our motivation is strongly rooted in industrial scenarios where business objectives frequently change. Compared to retraining models upon receiving new business goals, PadiRec **eliminates retraining costs** and **shortens the response time** from receiving a new objective **to conducting a new model**, as shown in Table 2 of our paper.
>
> [1] Neural network diffusion. _arXiv preprint arXiv:2402.13144_ (2024).
>
> [2] Spatio-temporal few-shot learning via diffusive neural network generation. ICLR, 2024
>
> # W3: SOTA Backbone
>
>
> Thanks for reminding me about the optimal backbone. It's important to clarify that PadiRec does **not focus on improving the accuracy** of sequential recommendation models. Instead, it provides a framework that can adapt to any downstream recommendation model, **with the goal of enabling customized recommendation models based on requirements without the need for retraining.** Nevertheless, we have supplemented our experiments with the SOTA backbone [1] and will include the relevant references and results in the main text. The experimental results are shown below. The trend figure with the backbone LRURec has been added in **Appendix § A.7 SOTA Backbone (LRURec).**
>
>
> | Backbone | Algorithm | Avg.HV | Pearson r-a | Pearson r-d |
> |----------|-----------|--------|--------------|-------------|
> | LRURec   | Retrain   | **0.2205** | -            | -           |
> | LRURec   | CMR       | 0.1887 | *0.9416 | *0.9551 |
> | LRURec   | Soup      | 0.1459 | 0.7902       | 0.8616      |
> | LRURec   | MMR       | 0.1911 | 0.8690       | 0.8043      |
> | LRURec   | PadiRec   | *0.2126 | **0.9977**  | **0.9983**  |
> | -        | LLM       | 0.0625 | -0.0922      | 0.09499     |
>
>
> [1] Linear recurrent units for sequential recommendation. WSDM, 2024.

---

> > ### Author Response · Authors · 2024-11-19
> >
> > # W4: Adding Fairness metrics
> > Thank you for your valuable suggestions. We have added user group fairness as a controllable metric. Specifically, we use the NDCG GAP@10 between male and female groups as the metric (a smaller value indicates greater fairness in NDCG@10 between the two groups) to evaluate the impact of fairness weights on the metric under different settings.
> >
> > Given the large number of possible weight combinations for the three objectives, we explored the impact of fairness through a controlled variable approach:
> >
> > 1. Investigating the impact of fairness weight on other metrics (NDCG@10 and a-NDCG@10).
> >
> > 2. Examining the effect of fairness weight on its metric (NDCG-GAP@10).
> >
> > The experimental results are presented below. The table records the performance of three metrics—accuracy (NDCG@10), diversity (a-NDCG@10), and fairness (NDCG-GAP@10)—under two conditions: **unfair** (fairness weight = 0.1) and **fair** (fairness weight = 1), as accuracy weight varies (constrained diversity weight = 1 - accuracy weight).
> >
> >
> > |       | Acc. weight | 0.0    | 0.1    | 0.2    | 0.3    | 0.4    | 0.5    | 0.6    | 0.7    | 0.8    | 0.9    | 1.0    |
> > |-------|-------------|--------|--------|--------|--------|--------|--------|--------|--------|--------|--------|--------|
> > | Fair  | NDCG@10     | 0.1062 | 0.1217 | 0.1288 | 0.1531 | 0.1760 | 0.2300 | 0.2959 | 0.3482 | 0.3814 | 0.4013 | 0.4072 |
> > |       | a-NDCG@10   | 0.1147 | 0.1151 | 0.1158 | 0.1147 | 0.1150 | 0.1136 | 0.1100 | 0.1027 | 0.0959 | 0.0854 | 0.0798 |
> > |       | NDCG_GAP    | 0.0234 | 0.0197 | 0.0161 | 0.0114 | 0.0036 | 0.0018 | 0.0119 | 0.0285 | 0.0197 | 0.0162 | 0.0245 |
> > | UnFair| NDCG@10     | 0.1048 | 0.1103 | 0.1185 | 0.1518 | 0.1800 | 0.2375 | 0.3034 | 0.3455 | 0.3807 | 0.4029 | 0.4054 |
> > |       | a-NDCG@10   | 0.1170 | 0.1168 | 0.1162 | 0.1164 | 0.1158 | 0.1130 | 0.1085 | 0.1019 | 0.0934 | 0.0844 | 0.0765 |
> > |       | NDCG_GAP@10    | 0.0236 | 0.0180 | 0.0127 | 0.0187 | 0.0045 | 0.0096 | 0.0267 | 0.0366 | 0.0313 | 0.0311 | 0.0293 |
> >
> > The line charts for the above metrics are presented in the PDF file, **Appendix § A.9 More Objectives (Accuracy, Diversity, and Fairness), Figure 15**. The following conclusions can be drawn:
> >
> > 1. Figures 15(a) and 15(b) show that the unfair and fair conditions have **minimal impact** on both **NDCG@10** and **a-NDCG@10** individually, as well as on the trade-off relationship between these two metrics.
> >
> > 2. Figure 15(c) demonstrates that the unfair and fair conditions **have an impact** on **NDCG-GAP@10**. Across multiple settings of accuracy weights, the NDCG-GAP@10 under the fair condition **is consistently smaller** than that under the unfair condition. This indicates that the control under the **fair/unfair** condition is effective.
> >
> > To explore the **fine-grained control** of fairness weight, we investigated the performance of PadiRec on the three objectives under fairness weights of 0.1, 0.4, 0.7, and 1.0 when accuracy weight is set to 0.6 and 0.7 (at these point, both NDCG@10 and alpha-NDCG@10 demonstrate not bad performance). The results are shown in the table below (we have added it to **Appendix § A.9 More Objectives (Accuracy, Diversity, and Fairness)**):
> >
> >
> > | Acc. weight | Fair. weight | NDCG@10 | a-NDCG@10 | NDCG_GAP@10 |
> > |-------------|--------------|---------|-----------|----------|
> > | 0.6         | 0.1          | 0.3034  | 0.1085    | 0.0267   |
> > |             | 0.4          | 0.2910  | 0.1096    | 0.0253   |
> > |             | 0.7          | 0.2945  | 0.1094    | 0.0175   |
> > |             | 1.0          | 0.2959  | 0.1100    | 0.0119   |
> > |-------------|--------------|---------|-----------|----------|
> > | 0.7         | 0.1          | 0.3455  | 0.1019    | 0.0366   |
> > |             | 0.4          | 0.3448  | 0.1019    | 0.0299   |
> > |             | 0.7          | 0.3395  | 0.1045    | 0.0286   |
> > |             | 1.0          | 0.3482  | 0.1027    | 0.0285   |
> >
> > Conclusion: As the **fairness weight increases**, accuracy (NDCG@10) and diversity (a-NDCG@10) show minimal fluctuation, while NDCG-GAP@10 **steadily decreases**, indicating improved fairness. This demonstrates that even under multiple objectives, PadiRec exhibits strong controllability.

---

> > > ### Author Response · Authors · 2024-11-19
> > >
> > > # Q1: Pearson r-d
> > > In this paper, "Pearson r-d" refers to the Pearson correlation coefficient between the retraining method and other methods with respect to the variable "diversity". The Pearson correlation coefficient is commonly used to **measure the correlation between two variables** and has been widely applied in previous works [1]. Given that Controllable Multi-Task Recommendation (CMTR) tasks **focus on controllability on multiple objectives**, and there is a significant lack of research in this area, we borrowed the concept of Hypervolume from Multi-Task Recommendation (MTR) tasks to measure the overall performance across multiple objectives. As for **controllability**, we expect the algorithm's performance under given preference weights to closely resemble that of the optimal (i.e., retrain on those preference weights) approach. Therefore, we use the Pearson correlation coefficient as a metric to assess controllability. This is an area that still requires further improvement, and we look forward to the development of more effective and refined metrics in the future.
> > >
> > > [1] Demystifying Causal Features on Adversarial Examples and Causal Inoculation for Robust Network by Adversarial Instrumental Variable Regression. CVPR, 2023.
> > >
> > > [2] Analyzing and Evaluating Correlation Measures in NLG Meta-Evaluation. arxiv, 2024.
> > >
> > > # Q2: Details of DIffusion Training
> > > This is a key point. In calculating the reported efficiency, the diffusion training time is not included in the results. This is because the time reported here refers to the time **from receiving a new requirement** (i.e., new preference weights) **to**the completion of the **model construction**. The diffusion model in **Padirec** requires only a single training session before deployment, with **no additional training** needed during the inference process. After deployment, Padirec can **directly generate** a customized model based on the preference weights. In contrast, traditional optimization methods usually require **retraining** the model from scratch according to the new preference weights, which is quite a time and computation-consuming.

---

> > > > ### Comment · Reviewer_kFtK · 2024-11-26
> > > >
> > > > Thank you to the authors for their diligent response, which addressed some of my concerns. However, I still have reservations regarding the novelty of the applied work in the ICLR community and the feasibility of the offline evaluation. Consequently, I will maintain my original ratings.

---

> ### Author Response · Authors · 2024-11-27
> **Regarding the ICLR topics and offline evaluation**
>
> Thanks for your response.
>
> The ICLR community allows, encourages, and supports applied research papers. For instance, ICLR 2025 explicitly lists topics including, but not limited to:
>
> > + Applications in audio, speech, robotics, neuroscience, biology, or any other field.
>
> Examples of applied works include those focusing on recommendation systems and retrieval [1, 2, 3, 4, 5], as well as studies effectively integrating diffusion models with various subfields based on their unique characteristics [6, 7, 8, 9]. Additionally, other trending applied research directions, such as LLM agents [10, 11, 12], are also actively encouraged by ICLR.
>
> It is crucial to emphasize that our paper focuses on **controllable multi-task recommendation**, specifically addressing **instant responsiveness and control** based on **dynamic changes in recommendation objectives.** Parameter diffusion aligns perfectly with this goal, enabling **test-time model customization** via the "one instruction, one model" paradigm. Furthermore, **conducting only offline evaluations** on widely recognized datasets for recommendation algorithms is a standard practice [1, 2, 4, 5]. Moreover, the third dataset used in this study—industrial data—is an **online dataset** collected from a real-world production environment, further validating the practical value of our method.
>
> Therefore, both in terms of **topic** and **evaluation**, this paper is **well-aligned with ICLR's scope** and standards. If there are additional technical concerns, we will do our best to address them.
>
>
>
> [1] Towards Unified Multi-Modal Personalization: Large Vision-Language Models for Generative Recommendation and Beyond. ICLR, 2024
>
> [2] Federated Recommendation with Additive Personalization. ICLR, 2024
>
> [3] Sentence-level Prompts Benefit Composed Image Retrieval. ICLR, 2024
>
> [4] LightGCL: Simple Yet Effective Graph Contrastive Learning for Recommendation. ICLR, 2023
>
> [5] StableDR: Stabilized Doubly Robust Learning for Recommendation on Data Missing Not at Random. ICLR, 2023
>
> [6] Spatio-Temporal Few-Shot Learning via Diffusive Neural Network Generation. ICLR, 2024
>
> [7] DiffAR: Denoising Diffusion Autoregressive Model for Raw Speech Waveform Generation. ICLR, 2024
>
> [8] Multi-Source Diffusion Models for Simultaneous Music Generation and Separation. ICLR, 2024
>
> [9] Training-free Multi-objective Diffusion Model for 3D Molecule Generation. ICLR, 2024
>
> [10] AutoGen: Enabling Next-Gen LLM Applications via Multi-Agent Conversation. ICLR, 2024
>
> [11] ChatEval: Towards Better LLM-based Evaluators through Multi-Agent Debate. ICLR, 2024
>
> [12] Plug-and-Play Policy Planner for Large Language Model Powered Dialogue Agents. ICLR,2024

---

> > ### Author Response · Authors · 2024-12-03
> >
> > Dear Reviewer,
> >
> > We kindly inquire whether our responses have adequately addressed your concerns. If there are any remaining misunderstandings or uncertainties, we would greatly value the opportunity to discuss and clarify them further with you.
> >
> > Best regards,
> > The Authors

---

### Official Review · Reviewer_oQdP · 2024-11-01

**Soundness:** 2
**Presentation:** 3
**Contribution:** 2
**Rating:** 3
**Confidence:** 4

**Summary:**

This paper highlights the dynamic preferences to be considered when deploying recommender systems in practice. Traditional models, including multi-task learning methods, cannot effectively handle dynamic user preferences. Although retraining is a common approach, it is very time-consuming and resource-intensive. Therefore, they proposed a parameter generation method for controllable multi-task recommendation, which effectively generates task-specific model parameters using a generative model. Specifically, they proposed PaDiRec by formulating an objective function consistent with task-specific preference weights and then using adapter tuning to fine-tune model parameters. They trained a diffusion model to learn the conditional distribution of these optimized adapter parameters. Finally, they evaluated PaDiRec using different sequential recommendation backbones to produce recommendations.

**Strengths:**

(1) This paper studies an interesting research problem that is common in the field of recommender systems.
(2) The authors proposed a series of methods, including adapter modules, accuracy + diversity objective functions, adapter tuning, etc.
(3) The authors proved their point through a large number of experiments

**Weaknesses:**

(1) The evaluation presented in this paper raises some concerns. First, the authors assessed the sequential recommendation task using the MovieLens dataset. I would strongly advise caution, as the user behaviors reflected in the MovieLens data are not genuine viewing behaviors but rather rating behaviors. Many users rate movies they have not actually watched, often based on prior knowledge. A closer examination of the dataset reveals that some users can rate five or even ten movies within just a few minutes.  The rating time in MovieLens is not the watching time! Also see  [1] by Sun et.

(2) The hyperparameters may not have been fine-tuned carefully. While the MovieLens dataset does exhibit high sequential patterns, it is important to note that, as mentioned, the user behaviors are not reflective of actual viewing. The sequential patter of recommendation systems employed by MovieLens can be easily detected using models like SASRec. I observed that the authors used an embedding size of 64; to my knowledge, SASRec's optimal embedding size is significantly larger than this in MovieLens. Utilizing a non-optimal embedding size for baseline models undermines the claim that the proposed model is superior. It’s essential to recognize that in deep learning, simply comparing models with the same embedding size is not a fair approach. Some deep learning models can effectively utilize larger embedding sizes, while others may reach saturation with smaller embedding sizes.

(3) Multi-task learning is often applied in the ranking stage, while sequential recommendation typically functions as a recall algorithm. I do not think it makes much sense to achieve better performance in the sequential recommendation task.

(4) In fact, having a more powerful sequential recommendation model may not be that important, as such improvements generally do not bring any benefits in the ranking stage or in online systems. Many sequential patterns learned by deep Transformer models are more closely related to recommendation exposure patterns than to actual user behavior patterns. Therefore, I do not think that the solution proposed in this paper has a particular impact on the current recommender system landscape.

[1] Our Model Achieves Excellent Performance on MovieLens: What Does It Mean? TOIS2024

**Questions:**

no

---

> ### Author Response · Authors · 2024-11-19
>
> Thank you for taking the time to review our paper, especially given your busy schedule. Your suggestions are highly detailed, and your perspectives are insightful! However, some parts of your feedback reflect  misunderstandings of our work. We have provided detailed explanations for each of your concerns and hope to clarify any misconceptions.
>
>
>
> # W1: Evaluation of Movielens
> Valuable suggestions! The paper you mentioned [1] by Sun et al. reveals an interesting phenomenon and highlights some **limitations of the MovieLens** dataset. However, it was **published on (October 18, 2024)** **after the submission deadline for ICLR 2025 (October 1, 2024)**. Therefore, the insights from this work cannot be considered as a reference for our ICLR submission.
>
> Additionally, many classic sequential recommendation papers [2, 3, 4, 5] have included experiments based on the MovieLens dataset. Among them, the methods **[2, 3]serve as our backbones.** To **align with these studies**, we also adopt the MovieLens dataset. Besides, this paper also presents experiments on the **Amazon dataset and real-world industrial datasets,** further demonstrating the generalizability of PadiRec. However, we will take your feedback regarding the MovieLens evaluation into account in our future work. Thanks again for your **valuable** suggestions!
>
> [1] Our Model Achieves Excellent Performance on MovieLens: What Does It Mean? TOIS, 2024
>
> [2] Self-Attentive Sequential Recommendation. ICDM, 2018
>
> [3] TiSASRec: Time Interval Aware Self-Attention for Sequential Recommendation. WSDM, 2020
>
> [4] BERT4Rec: Sequential Recommendation with Bidirectional Encoder Representations from Transformer. CIKM, 2019
>
> [5] Generative Sequential Recommendation with GPTRec. SIGIR, 2023

---

> > ### Author Response · Authors · 2024-11-19
> >
> > # W2: Embedding Size:
> > Our primary motivation is **controllability**. Simply improving the accuracy of recommendation models is not the primary goal of this work. Widely recognized frameworks like RecBole [1] set the default embedding size to 64. In fact, we maintain a consistent embedding size across all models to **ensure fairness** among baselines, as different embedding sizes would **alter the adapter dimensions**, potentially **impacting the diffusion model's ability** to learn the adapter parameters effectively. Nevertheless, based on your suggestion, we conducted additional experiments using SASRec as the backbone of the Movielens dataset with other embedding sizes (embedding size = 128, 256). The experimental results are presented below.
> >
> > | Emb. size | Acc. weight | 0.0     | 0.1     | 0.2     | 0.3     | 0.4     | 0.5     | 0.6     | 0.7     | 0.8     | 0.9     | 1.0     |
> > |-----------|-------------|---------|---------|---------|---------|---------|---------|---------|---------|---------|---------|---------|
> > |           | **Acc.(NDCG@10)**                                                                                                     |
> > | 64        |             | 0.0857  | 0.1327  | 0.1643  | 0.1917  | 0.2276  | 0.2717  | 0.3204  | 0.3656  | 0.3945  | 0.4083  | 0.4165  |
> > | 128       |             | 0.0961  | 0.1137  | 0.1328  | 0.1596  | 0.1911  | 0.2373  | 0.3013  | 0.3541  | 0.3841  | 0.4066  | 0.4154  |
> > | 256       |             | 0.0883  | 0.1108  | 0.1275  | 0.1593  | 0.1967  | 0.2482  | 0.3232  | 0.3694  | 0.3943  | 0.4158  | **0.4174** |
> > |           | **Div.(a-NDCG@10)**                                                                                                |
> > | 64        |             | 0.1184  | 0.1175  | 0.1169  | 0.1161  | 0.1152  | 0.1129  | 0.1087  | 0.1022  | 0.0940  | 0.0856  | 0.0787  |
> > | 128       |             | 0.1186  | 0.1193  | 0.1192  | 0.1189  | 0.1182  | 0.1162  | 0.1117  | 0.1043  | 0.0960  | 0.0871  | 0.0779  |
> > | 256       |             | **0.1221** | 0.1221  | 0.1222  | 0.1217  | 0.1206  | 0.1183  | 0.112   | 0.1036  | 0.0952  | 0.0858  | 0.0773  |
> >
> > When training focuses solely on accuracy (i.e., acc. weight = 1.0), NDCG@10 achieves its highest value when the embedding size is set to 256. Similarly, when training focuses solely on diversity (i.e., div. weight = 1.0), a-NDCG@10 reaches its peak at an embedding size of 256. These results suggest that the current setting of embedding size = 64 has room for improvement. Based on these embedding sizes, we conducted subsequent diffusion-based conditional training and generation. The results are shown below. The figure has been added in  **Appendix § A.8 The Embedding Size Problem.**
> >
> >
> > | Backbone        | Algorithm | Avg.HV | Pearson r-a | Pearson r-d |
> > |-----------------|-----------|--------|--------------|-------------|
> > | 128 emb_size SASRec   | Retrain   | 0.2160 | -            | -           |
> > | 128 emb_size SASRec   | CMR       | 0.2132 | 0.9676       | *0.9782 |
> > | 128 emb_size SASRec  | Soup      | *0.2228 | *0.9824 | 0.9626 |
> > | 128 emb_size SASRec | MMR       | 0.1802 | 0.8958       | 0.9031      |
> > | 128 emb_size SASRec   | PadiRec   | **0.2334** | **0.9971** | **0.9970** |
> > | -               | LLM       | 0.0625 | -0.1028      | 0.0805      |
> >
> >
> > | Backbone             | Algorithm | Avg.HV  | Pearson r-a | Pearson r-d |
> > |----------------------|-----------|---------|-------------|-------------|
> > | 256 emb_size SASRec  | Retrain   | 0.2214  | -           | -           |
> > | 256 emb_size SASRec  | CMR       | 0.2120  | 0.9504      | 0.9610  |
> > | 256 emb_size SASRec  | Soup      | **0.2286** | *0.9833  | *0.9680      |
> > | 256 emb_size SASRec  | MMR       | 0.1793  | 0.9133      | 0.8457      |
> > | 256 emb_size SASRec  | PadiRec   | *0.2262 | **0.9926** | **0.9959** |
> > | -                    | LLM       | 0.0625  | -0.0754     | 0.0754      |
> >
> >
> >
> > [1] Towards a More User-Friendly and Easy-to-Use Benchmark Library for Recommender System. SIGIR, 2023.

---

> > > ### Author Response · Authors · 2024-11-19
> > >
> > > # W3: Sequential Recommendation Task is Crucial for Multi-task Learning.
> > > Recall is the foundation of ranking tasks. For example, **neglecting diversity (and other factors) during recall** can result in a more homogeneous set of candidates, which inherently **limits the upper bound** of diversity achievable in ranking.
> > >
> > > On the other hand, sequential recommendation models indeed could serve both recall and ranking. Many **research works** distinguish the two primarily based on whether the candidates in the experimental setting are a **universal set** or a **subset**. Notably, numerous studies adopt this setting and utilize sequential recommendation models for ranking tasks [1, 2, 3].
> > >
> > > What's more, Padirec is a framework designed to enhance the controllability of downstream models. Our focus is on controlling the downstream models rather than on a specific recommendation model itself.
> > >
> > > [1] gSASRec: Reducing Overconfidence in Sequential Recommendation Trained with Negative Sampling. RecSys, 2023
> > >
> > > [2] Self-Attentive Sequential Recommendation. ICDM, 2018
> > >
> > > [3] GRU4Rec: Session-based Recommendations with Recurrent Neural Networks. ICLR, 2016
> > >
> > > [4] Large Language Models are Zero-Shot Rankers for Recommender Systems. ECIR, 2024
> > >
> > > # W4: We Make Recommendation Model Not More Powerful But Controllable
> > > We quite agree that having a more powerful sequential recommendation model may not be that important. In fact, our focus is **not on endlessly improving the recommendation accuracy**, but rather on **expanding the controllability** to the downstream recommendation model. That is, enabling the platform to avoid repeatedly retraining recommendation models based on frequently changing business metrics,and instead, shifting towards a task where models can be quickly generated with the required capabilities based on specific instructions. This paradigm offers several advantages, such as eliminating training costs and accelerating the response time to new instructions. This is a bold attempt on our part, and we are fortunate that it aligns with your thoughts.

---

> ### Comment · Reviewer_oQdP · 2024-11-26
> **Thanks**
>
> I have reviewed the authors' response, but I still have concerns. The problem like using MovieLens's for sequential recommendation - I strongly recommend consulting the creators of MovieLens to understand why it is not suitable for sequential recommendation tasks. I am not convinced by using the existing literature. The Recsys models were not properly evaluated in the much literature.
> More broadly, I do not see how this work significantly advances the field given its overall contribution. While I respect the editor's final decision, I stand by my original score because I do not think the paper's contribution is sufficient to be accepted in ICLR.

---

> > ### Author Response · Authors · 2024-11-26
> > **Concerns regarding the dataset and the topic.**
> >
> > Thank you for your response. We believe there are still some misunderstandings about our paper.
> >
> > # Regarding the dataset:
> >
> > **First,** we must reiterate that the article discussing the limitations of MovieLens [1] was **published after our submission.**
> >
> > **Second,** nonetheless, we have carefully reviewed the article [1]. In fact, our approach **aligns with** the final suggestions of this paper [1]. The original text from **§ 5.4 Is It a Good Idea to Evaluate RecSys Models on MovieLens**  is as follows:
> >
> > > On the other hand, MovieLens stands out as one of **the most popular datasets** in the field of recommender systems [2, 3, 4, 5], Results derived from MovieLens serve as valuable references for researchers to validate their implementations... In short, while **providing results on MovieLens for reference purposes is beneficial**, it should not serve as a strong justification for the effectiveness of a proposed model. Models should be evaluated on a variety of datasets, **not relying solely on the MovieLens dataset.**
> >
> > Our paper includes experiments **conducted on THREE datasets**, **not solely on MovieLens**, which **aligns with** the suggestions in the paper [1]. It **cannot be overlooked** that we conducted experiments on **THREE datasets**. If you believe there are additional datasets beyond the **THREE datasets**, please let us know, and we will do our best to address your concerns.
> >
> > Finally, the discussion and statement regarding the MovieLens evaluation **have been added to the PDF file in § Appendix A.15 Discussion on used Movielens Evaluation.**  However, it is **important to note** again, **our focus** is **not on the patterns of sequential data** but rather on **providing controllability** to the backbone models.
> >
> > # Regarding the topic of the paper:
> >
> > The sequential recommendation model serves only as a backbone in our work. The goal of our algorithm is to enhance the **controllability** of **any given backbone**. More specifically, our paper primarily **addresses** the challenge of **inefficient retraining of deployed models at test time,** thereby improving the models' ability to **adapt dynamically to changes in task requirements**.
> >
> >
> >
> > [1] Our Model Achieves Excellent Performance on MovieLens: What Does It Mean? TOIS, 2024
> >
> > [2] Self-Attentive Sequential Recommendation. ICDM, 2018
> >
> > [3] TiSASRec: Time Interval Aware Self-Attention for Sequential Recommendation. WSDM, 2020
> >
> > [4] BERT4Rec: Sequential Recommendation with Bidirectional Encoder Representations from Transformer. CIKM, 2019
> >
> > [5] Generative Sequential Recommendation with GPTRec. SIGIR, 2023

---

> ### Comment · Reviewer_oQdP · 2024-11-27
> **Thanks**
>
> "First, we must reiterate that the article discussing the limitations of MovieLens [1] was published after our submission."
>
> Even without this paper, I still believe that researchers in this field should know what kind of tasks MovieLens can do. It is a rating prediction dataset, where uses were paid to rate movies. Many users can rate over 10 movies in 1-2 minutes.  Rating sequences should not be considered as user real watching behavior sequences. The sequential patterns you captured come from the original MovieLens exposure strategy.
>
> The following comments only represent personal opinions, and you can ignore them if you disagree.
>
> 1）Sequential recommendation on offline datasets has inherent limitations. The patterns we observe are largely artifacts of the original recommendation algorithms rather than true user behavior, which rarely follows strict sequences. This task was usually called session-based recommendation in the past, which acknowledges that user preferences tend to remain stable within short time windows.
>
> 2）Further improvements in offline accuracy metrics offer diminishing returns for these sequential task. While I understand academic researchers' limited access to online datasets, the field has reached established in terms of offline performance metrics.
>
> Overall, I don’t think this paper is important to the Recsys community.

---

> > ### Author Response · Authors · 2024-11-27
> > **Regarding dataset and our focus**
> >
> > Thank you for your feedback.
> >
> > We appreciate your concerns regarding the collection methods of public datasets. However, we believe this discussion somewhat **exceeds the scope of our work**. We would like to reiterate that this study primarily focuses on controllable multi-task recommendation, rather than sequential recommendation. The user behavior sequences are included in experiments solely as user features, and our **focus lies on the adaptive adjustment and control of model parameters** in response to **dynamic changes in recommendation objectives** based on parameter diffusion. The exploration of user sequence behavior patterns is not within the scope of this study, and the user behavior sequences could be replaced by other user features (e.g., collaborative filtering features).
> >
> > Additionally, regarding offline evaluation, we **do not merely emphasize "improving offline accuracy metrics"** as the reviewer noted. Instead, we argue that at test time, improving accuracy alone is insufficient. It is crucial to enable the model to **adapt to dynamic changes in platform or user requirements across multiple metrics,** which is the **core topic of this work**. Offline testing is an essential step for any model being deployed in practice, as it is widely recognized to correlate positively with online performance. Furthermore, as the reviewer suggested, the **third dataset used in this study**—industrial data—is an **online dataset** collected from a real-world product environment, further validating the practical value of our proposed method.
> >
> > Finally, classic sequential recommendation works [2, 3, 4, 5]  utilized MovieLens. Some submissions to ICLR 2025 [1] have used sequence models on MovieLens and received high scores. **Moreover, [6] points out the limitations of MovieLens but it advocates more datasets including Movielens.** Therefore, using MovieLens is not a valid reason for rejection. To reiterate, we employed **THREE(!) datasets** in our experiments. Our focus is not on improving accuracy but on **CONTROLLABILITY(!)**. We look forward to engaging in further technical discussions.
> >
> > Once again, thank you for your feedback. We welcome further academic discussions relevant to the theme of this study.
> >
> >
> >
> > [1] Preference Diffusion for Recommendation. under review ICLR,2025
> >
> > [2] Self-Attentive Sequential Recommendation. ICDM, 2018
> >
> > [3] TiSASRec: Time Interval Aware Self-Attention for Sequential Recommendation. WSDM, 2020
> >
> > [4] BERT4Rec: Sequential Recommendation with Bidirectional Encoder Representations from Transformer. CIKM, 2019
> >
> > [5] Generative Sequential Recommendation with GPTRec. SIGIR, 2023
> >
> > [6] Our Model Achieves Excellent Performance on MovieLens: What Does It Mean? TOIS, 2024

---

> > > ### Comment · Reviewer_oQdP · 2024-11-27
> > > **see below**
> > >
> > > “Finally, classic sequential recommendation works [2, 3, 4, 5] utilized MovieLens. ”
> > >
> > > In the field of Recsys, there are a lot of inappropriate evaluations even we only consider the offline settings (see Recsys or SIGIR's reproducibility track). Using MovieLens for sequential recommendation is a VERY big problem. I have explained the reasons above.
> > >
> > >  "the third dataset used in this study—industrial data—is an online dataset collected from a real-world product environment, further validating the practical value of our proposed method."
> > >
> > > I have not seen your description about the online setting like A/B test. It seems that the dataset is still an offline dataset, but from an industry system. Your Amazon is also an industrial dataset. Using an Industrial datasets does not mean you evaluate your model on the online setting.
> > >
> > >
> > > BTW: I did not find details about how you split the data into training and testing sets. Please clarify your data partitioning strategy.

---

> > > > ### Author Response · Authors · 2024-11-27
> > > > **Our focus and dataset setting**
> > > >
> > > > Tank you for your response. Our algorithm is compatible with any recommendation model and aims to **provide controllability for downstream models**. Specifically, given a requirement, it can quickly respond and **customize a recommendation model** to **meet that requirement**. The user behavior sequences are included in the experiments solely as user features and are not our focus. While we acknowledge the limitations of MovieLens, they do not affect the core goal of our algorithm: **controlling downstream models**. Furthermore, the other two datasets consist of **real click behavior data** (unlike the rating data in MovieLens), which effectively demonstrate the applicability of our method.
> > > >
> > > > For data processing, we used the ReChorus2.0 framework (https://github.com/THUwangcy/ReChorus) to standardize the data. The data was split using the Leave-One-Out approach, and detailed information has been updated in the **PDF file** **Appendix § A 3.2 Dataset Settings.**

---

> > > > ### Author Response · Authors · 2024-11-28
> > > > **Follow-up on Concerns**
> > > >
> > > > Dear Reviewer,
> > > >
> > > > We kindly ask if our responses have addressed your concerns. If there are still any misunderstandings or uncertainties, we would greatly appreciate the opportunity to further discuss and clarify them with you.
> > > >
> > > > Best,
> > > > Authors

---

> > > > > ### Comment · Reviewer_oQdP · 2024-12-03
> > > > > **I keep the original score**
> > > > >
> > > > > I appreciate the authors' response, but I do not think my key concerns have been solved. Overall, I evaluated this work based on how it advances the field.

---

> > > > > > ### Author Response · Authors · 2024-12-03
> > > > > > **Summary**
> > > > > >
> > > > > > Dear Reviewer,
> > > > > >
> > > > > > It seems there is still a misunderstanding, and we would like to summarize our points here in the hope of providing a clearer understanding.
> > > > > >
> > > > > > 1. **The reviewer believes that the MovieLens dataset is unsuitable for sequential recommendation.**
> > > > > >
> > > > > >    - **Our opinion:**
> > > > > >       - In fact, we have validated the effectiveness of our algorithm **across multiple datasets**, not just MovieLens.
> > > > > >       - The paper discussing the limitations of MovieLens was published after our ICLR submission.
> > > > > >       - As the reviewer mentioned, this issue pertains to the broader field of Recommender Systems. The MovieLens dataset has been widely used in many classic works on sequential recommendation. We are just following their settings widely adopted in the field to provide **a fair and consistent reference**.
> > > > > >
> > > > > > 2. **The reviewer believes that improving offline accuracy metrics has diminishing returns.**
> > > > > >
> > > > > >    - **Our opinion:**
> > > > > >      - Our primary contribution is not merely improving accuracy but rather enhancing **controllability**.
> > > > > >        We argue that improving accuracy alone at test time is insufficient. Instead, it is critical to enable the model to adapt dynamically to changes in platform or user requirements across multiple metrics (e.g., **diversity, accuracy, fairness**). This adaptability is the core focus of our work.
> > > > > >
> > > > > > In summary, we believe that the reviewer’s concerns about the dataset and the evaluation of the entire field may overshadow our primary contribution to advancing controllability in the recommendation community.
> > > > > >
> > > > > > Best regards,
> > > > > > The Authors

---

### Official Review · Reviewer_gFtT · 2024-11-03

**Soundness:** 3
**Presentation:** 3
**Contribution:** 3
**Rating:** 6
**Confidence:** 2

**Summary:**

The paper introduces PaDiRec, Parameter Diffusion for Controllable Multi-Task Recommendation, a framework designed to adapt recommender systems to changing task requirements without retraining. Traditional recommendation models often struggle to adjust dynamically. PaDiRec addresses this issue using a parameter diffusion model that can modify model parameters at test time, making it a model-agnostic solution adaptable to various backbone structures. Experiments on public datasets and an industrial dataset show that PaDiRec achieves high controllability and performance.

**Strengths:**

PaDiRec enables real-time, on-demand changes in task preferences without requiring expensive retraining, making it ideal for industry. The use of diffusion models as parameter generators is also novel in the context of recommender systems, offering robust parameter generation that captures task-specific nuances.

The paper provides well-designed experiments and several datasets. According to the results, PaDiRec integrates well with diverse backbone models, proving its applicability across different recommendation systems and settings as well as the capability to achieve near real-time responses with significantly reduced latency compared to traditional retraining.

**Weaknesses:**

The approach relies heavily on predefined utilities which may not fully generalize to tasks with complex, non-linear objectives. This dependency could limit its flexibility in handling multi-faceted objectives beyond accuracy and diversity.

While PaDiRec performs faster than retraining, the diffusion process may still be computationally intensive, especially in environments with limited processing power. More details on efficiency optimization could enhance its feasibility for larger-scale systems.

**Questions:**

How should practitioners choose between the various conditioning strategies (e.g., Pre&Post, Ada-Norm) when applying PaDiRec to new recommendation environments? Are there guidelines or heuristics to assist in selecting the optimal strategy?

How would PaDiRec handle non-standard or non-linear preference metrics? Would additional diffusion model tuning be required?

---

> ### Author Response · Authors · 2024-11-19
>
> Thank you for your time, effort, and valuable suggestions! We have provided detailed explanations and supplemented the experiments based on your suggestions. We hope this is helpful to you, and once again, we sincerely appreciate your effort in reviewing our paper despite your busy schedule.
>
> # W1: More Utilities:
> Thank you for your valuable suggestions. We have added user group fairness as a controllable metric. Specifically, we use the NDCG GAP@10 between male and female groups as the metric (a smaller value indicates greater fairness in NDCG@10 between the two groups) to evaluate the impact of fairness weights on the metric under different settings.
>
> Given the large number of possible weight combinations for the three objectives, we explored the impact of fairness through a controlled variable approach:
>
> 1. Investigating the impact of fairness weight on other metrics (NDCG@10 and a-NDCG@10).
>
> 2. Examining the effect of fairness weight on its metric (NDCG-GAP@10).
>
> The experimental results are presented below. The table records the performance of three metrics—accuracy (NDCG@10), diversity (a-NDCG@10), and fairness (NDCG-GAP@10)—under two conditions: **unfair** (fairness weight = 0.1) and **fair** (fairness weight = 1), as accuracy weight varies (constrained diversity weight = 1 - accuracy weight).
>
>
> |       | Acc. weight | 0.0    | 0.1    | 0.2    | 0.3    | 0.4    | 0.5    | 0.6    | 0.7    | 0.8    | 0.9    | 1.0    |
> |-------|-------------|--------|--------|--------|--------|--------|--------|--------|--------|--------|--------|--------|
> | Fair  | NDCG@10     | 0.1062 | 0.1217 | 0.1288 | 0.1531 | 0.1760 | 0.2300 | 0.2959 | 0.3482 | 0.3814 | 0.4013 | 0.4072 |
> |       | a-NDCG@10   | 0.1147 | 0.1151 | 0.1158 | 0.1147 | 0.1150 | 0.1136 | 0.1100 | 0.1027 | 0.0959 | 0.0854 | 0.0798 |
> |       | NDCG_GAP    | 0.0234 | 0.0197 | 0.0161 | 0.0114 | 0.0036 | 0.0018 | 0.0119 | 0.0285 | 0.0197 | 0.0162 | 0.0245 |
> | UnFair| NDCG@10     | 0.1048 | 0.1103 | 0.1185 | 0.1518 | 0.1800 | 0.2375 | 0.3034 | 0.3455 | 0.3807 | 0.4029 | 0.4054 |
> |       | a-NDCG@10   | 0.1170 | 0.1168 | 0.1162 | 0.1164 | 0.1158 | 0.1130 | 0.1085 | 0.1019 | 0.0934 | 0.0844 | 0.0765 |
> |       | NDCG_GAP@10    | 0.0236 | 0.0180 | 0.0127 | 0.0187 | 0.0045 | 0.0096 | 0.0267 | 0.0366 | 0.0313 | 0.0311 | 0.0293 |
>
>
> The line charts for the above metrics are presented in the PDF file, **Appendix § A.9 More Objectives (Accuracy, Diversity, and Fairness), Figure 15**. The following conclusions can be drawn:
>
> 1. Figures 15(a) and 15(b) show that the unfair and fair conditions have **minimal impact** on both **NDCG@10** and **a-NDCG@10** individually, as well as on the trade-off relationship between these two metrics.
>
> 2. Figure 15(c) demonstrates that the unfair and fair conditions **have an impact** on **NDCG-GAP@10**. Across multiple settings of accuracy weights, the NDCG-GAP@10 under the fair condition **is consistently smaller** than that under the unfair condition. This indicates that the control under the **fair/unfair** condition is effective.
>
> To explore the **fine-grained control** of fairness weight, we investigated the performance of PadiRec on the three objectives under fairness weights of 0.1, 0.4, 0.7, and 1.0 when accuracy weight is set to 0.6 and 0.7 (at these point, both NDCG@10 and alpha-NDCG@10 demonstrate not bad performance). The results are shown in the table below (we have added it to **Appendix § A.9 More Objectives (Accuracy, Diversity, and Fairness)**):
>
>
>
> | Acc. weight | Fair. weight | NDCG@10 | a-NDCG@10 | NDCG_GAP@10 |
> |-------------|--------------|---------|-----------|----------|
> | 0.6         | 0.1          | 0.3034  | 0.1085    | 0.0267   |
> |             | 0.4          | 0.2910  | 0.1096    | 0.0253   |
> |             | 0.7          | 0.2945  | 0.1094    | 0.0175   |
> |             | 1.0          | 0.2959  | 0.1100    | 0.0119   |
> |-------------|--------------|---------|-----------|----------|
> | 0.7         | 0.1          | 0.3455  | 0.1019    | 0.0366   |
> |             | 0.4          | 0.3448  | 0.1019    | 0.0299   |
> |             | 0.7          | 0.3395  | 0.1045    | 0.0286   |
> |             | 1.0          | 0.3482  | 0.1027    | 0.0285   |
>
> Conclusion: As the **fairness weight increases**, accuracy (NDCG@10) and diversity (a-NDCG@10) show minimal fluctuation, while NDCG-GAP@10 **steadily decreases**, indicating improved fairness. This demonstrates that even under multiple objectives, PadiRec exhibits strong controllability.

---

> > ### Author Response · Authors · 2024-11-19
> >
> > # W2: Inference Cost of Diffusion
> > As shown in Table 2, the diffusion model significantly reduces the time overhead, from receiving a new task instruction to completing the model construction, compared to traditional retraining approaches.
> >
> > As for the inference cost of diffusion, Our algorithm design has taken this into account from two aspects. One is the design to structure the recommendation model as a **backbone + adapter architecture****.** The diffusion model is used solely to generate the adapter parameters, with the goal of reducing the training and generation overhead of diffusion. The other is the design of the denoising model of diffusion. We **only stack** **4** **attention** to act as a denoising model. The details of the diffusion model have been added to the PDF file and can be found in **Appendix § A.12, Details of Diffusion**. Additionally, we have included a comprehensive analysis of the computational cost, provided in **Appendix § A.13, Diffusion Transformer FLOPs Calculation.**
> >
> > The conclusion of **Appendix § A.13, shows** that our diffusion model requires **only 0.9085 TFLOPs for the entire 500-step** sampling process. Using the RTX 3090 as an example, which achieves 35.58 TFLOPS per second, the inference process for the diffusion model takes approximately **0.026 seconds**. Including some data storage overhead, the total time remains **within the order of seconds** (as shown in Table 2, about 2.68s of SASRec). In real-world recommendation scenarios, a single recommendation typically occurs within milliseconds. However, waiting 2-3 seconds to customize a more personalized model is generally **acceptable** for users.
> >
> > # Q1: Conditioning Strategies
> > In fact, our experiments (Figure. 5) show that while there are differences in performance across different strategies, these differences are not significant. When applied to other recommendation scenarios, there may be no need to differentiate between strategies, or they could be treated as hyperparameters to be selected through experimentation.

---

> ### Comment · Reviewer_gFtT · 2024-11-29
>
> Thank you for addressing my comments and concerns in your rebuttal. I appreciate the effort you put into clarifying my doubts and putting user group fairness into your metrics, which in my mind have made the paper more robust.
>
> I want to emphasize that my initial score already reflected my positive view of the paper's quality and potential. While the rebuttal has resolved my concerns and improved the paper, there were no substantial changes that would warrant increasing the score further. As such, I have decided to maintain my original evaluation. Thank you for authors contributions to ICLR community.

---

> > ### Author Response · Authors · 2024-11-30
> >
> > Dear Reviewer,
> >
> > Thank you very much for your thoughtful feedback and for recognizing our efforts in addressing your comments and concerns. We appreciate your positive view of our work, and we are glad that our responses clarified your doubts and made the paper more robust.
> >
> > We would like to highlight that we carefully addressed each of your concerns, and we did not find specific requests for "substantial changes" in your initial review. Nonetheless, we remain committed to making further improvements if needed, and we look forward to any further discussion or suggestions you may have.
> >
> > Thank you again for your time and contributions to the ICLR community.
> >
> > Best regards

---

### Official Review · Reviewer_8b4P · 2024-11-05

**Soundness:** 3
**Presentation:** 2
**Contribution:** 3
**Rating:** 6
**Confidence:** 3

**Summary:**

This paper proposes a new learning method called "parameter diffusion" for controllable multi-task recommendation, which allows customization and adaptation of recommendation model parameters to new task requirements without retraining. The proposed method uses existing optimized model parameters to generate new ones that are adapted to different task requirements through parameter diffusion. The main contribution of this paper is to provide a new learning method that enhances the controllability and flexibility of multi-task recommendation systems.

**Strengths:**

•	Tackles the practical issue of dynamic task requirements
•	Allows customization and adaptation of recommendation model parameters to new task requirements without retraining
•	Novel combination of diffusion models with adapter tuning

**Weaknesses:**

•	High inference complexity of diffusion models may not meet real-time recommendation requirements. No detailed analysis of computational overhead during inference

•	Parameter Optimization Strategy:
o	Relies solely on single-task optimized parameters, ignores potential benefits of multi-task joint optimization

•	Limited Generalization and Flexibility:
o	The method is primarily tested on only two specific tasks (accuracy and diversity); Unclear scalability to more utilities/tasks.
o	Predefined preference weights limit the model's adaptability to unexpected scenarios

**Questions:**

Are there any standard or guiding principles that can help us choose appropriate task requirements (preference weights)?

---

> ### Author Response · Authors · 2024-11-19
>
> Thank you very much for your recognition of our work! Here, we respond to each of your concerns regarding our study.
>
> # W1: Inference Complexity
> As shown in Table 2, the diffusion model significantly reduces the time overhead from receiving a new task instruction to completing the model construction, **compared to traditional retraining approaches**.
>
> Technically, our algorithm design has taken the inference cost into account from two aspects. One is the design to structure the recommendation model as a **backbone + adapter** architecture. The diffusion model is used solely to generate the adapter parameters, to reduce the training and generation overhead of diffusion. The other is the design of the denoising model of diffusion. We **only stack 4 attention** to act as a denoising model. The details of the diffusion model have been added to the PDF file and can be found in **Appendix § A.13, Details of Diffusion.** Additionally, we have **added** a comprehensive analysis of the computational cost, provided in **Appendix § A.14 Diffusion Transformer FLOPs Calculation.**
>
> The conclusion of Appendix § A.14shows that our diffusion model requires **only 0.9085 TFLOPs for the entire 500-step** sampling process. Using the RTX 3090 as an example, which achieves 35.58 TFLOPS per second, the inference process for the diffusion model takes approximately **0.026 seconds**. Including some data storage overhead, the total time remains **within the order of seconds** (as shown in Table 2, about 2.68s of SASRec). In real-world recommendation scenarios, it is generally **acceptable** for users to wait 2-3 seconds to customize a more personalized model.
>
> # W2: Joint Optimization
> In §2 Problem Formulation and Analysis, we define controllable multi-task recommendation (**CMTR**) and distinguish it from multi-task recommendation (MTR). Under the definition of CMTR, a single task is defined as **an optimization problem with a specific set of preference weights** for multiple objectives. Therefore, a "task" under CMTR inherently involves the joint optimization of multiple objectives.

---

> > ### Author Response · Authors · 2024-11-19
> >
> > # W3-1: Scalability to More Objectives
> > Thank you for your valuable suggestions. We have added user group fairness as a controllable metric. Specifically, we use the NDCG GAP@10 between male and female groups as the metric (a smaller value indicates greater fairness in NDCG@10 between the two groups) to evaluate the impact of fairness weights on the metric under different settings.
> >
> > Given the large number of possible weight combinations for the three objectives, we explored the impact of fairness through a controlled variable approach:
> >
> > 1. Investigating the impact of fairness weight on other metrics (NDCG@10 and a-NDCG@10).
> >
> > 2. Examining the effect of fairness weight on its metric (NDCG-GAP@10).
> >
> > The experimental results are presented below. The table records the performance of three metrics—accuracy (NDCG@10), diversity (a-NDCG@10), and fairness (NDCG-GAP@10)—under two conditions: **unfair** (fairness weight = 0.1) and **fair** (fairness weight = 1), as accuracy weight varies (constrained diversity weight = 1 - accuracy weight).
> >
> >
> > |       | Acc. weight | 0.0    | 0.1    | 0.2    | 0.3    | 0.4    | 0.5    | 0.6    | 0.7    | 0.8    | 0.9    | 1.0    |
> > |-------|-------------|--------|--------|--------|--------|--------|--------|--------|--------|--------|--------|--------|
> > | Fair  | NDCG@10     | 0.1062 | 0.1217 | 0.1288 | 0.1531 | 0.1760 | 0.2300 | 0.2959 | 0.3482 | 0.3814 | 0.4013 | 0.4072 |
> > |       | a-NDCG@10   | 0.1147 | 0.1151 | 0.1158 | 0.1147 | 0.1150 | 0.1136 | 0.1100 | 0.1027 | 0.0959 | 0.0854 | 0.0798 |
> > |       | NDCG_GAP    | 0.0234 | 0.0197 | 0.0161 | 0.0114 | 0.0036 | 0.0018 | 0.0119 | 0.0285 | 0.0197 | 0.0162 | 0.0245 |
> > | UnFair| NDCG@10     | 0.1048 | 0.1103 | 0.1185 | 0.1518 | 0.1800 | 0.2375 | 0.3034 | 0.3455 | 0.3807 | 0.4029 | 0.4054 |
> > |       | a-NDCG@10   | 0.1170 | 0.1168 | 0.1162 | 0.1164 | 0.1158 | 0.1130 | 0.1085 | 0.1019 | 0.0934 | 0.0844 | 0.0765 |
> > |       | NDCG_GAP@10    | 0.0236 | 0.0180 | 0.0127 | 0.0187 | 0.0045 | 0.0096 | 0.0267 | 0.0366 | 0.0313 | 0.0311 | 0.0293 |
> >
> >
> > The line charts for the above metrics are presented in the PDF file, **Appendix § A.9 More Objectives (Fairness), Figure 15**. The following conclusions can be drawn:
> >
> > 1. Figures 15(a) and 15(b) show that the unfair and fair conditions have **negligible impact** on both **NDCG@10** and **a-NDCG@10** individually, as well as on the trade-off relationship between these two metrics.
> >
> > 2. Figure 15(c) demonstrates that the unfair and fair conditions **have an impact** on **NDCG_GAP@10**. Across multiple settings of accuracy weights, the NDCG-GAP@10 under the fair condition **is consistently smaller** than that under the unfair condition. This indicates that the control under the **fair/unfair** condition is effective.
> >
> > To explore the **fine-grained control** of fairness weight, we investigated the performance of PadiRec on the three objectives under fairness weights of 0.1, 0.4, 0.7, and 1.0 when accuracy weight is set to 0.6 and 0.7 (at these points, both NDCG@10 and alpha-NDCG@10 demonstrate not bad performance). The results are shown in the table below (we have added it to **Appendix § A.9 More Objectives (Accuracy, Diversity, and Fairness)**):
> >
> >
> > | Acc. weight | Fair. weight | NDCG@10 | a-NDCG@10 | NDCG_GAP@10 |
> > |-------------|--------------|---------|-----------|----------|
> > | 0.6         | 0.1          | 0.3034  | 0.1085    | 0.0267   |
> > |             | 0.4          | 0.2910  | 0.1096    | 0.0253   |
> > |             | 0.7          | 0.2945  | 0.1094    | 0.0175   |
> > |             | 1.0          | 0.2959  | 0.1100    | 0.0119   |
> > |-------------|--------------|---------|-----------|----------|
> > | 0.7         | 0.1          | 0.3455  | 0.1019    | 0.0366   |
> > |             | 0.4          | 0.3448  | 0.1019    | 0.0299   |
> > |             | 0.7          | 0.3395  | 0.1045    | 0.0286   |
> > |             | 1.0          | 0.3482  | 0.1027    | 0.0285   |
> >
> >
> > Conclusion: As the **fairness weight increases**, accuracy (NDCG@10) and diversity (a-NDCG@10) remain almost unchanged, while NDCG-GAP@10 **steadily decreases**, indicating improved fairness. This demonstrates that even under multiple objectives, PadiRec exhibits strong controllability.

---

> > > ### Author Response · Authors · 2024-11-19
> > >
> > > # W3-2: Preference Weights Limitation
> > > In  §2, we have limited the scope of the scenario to various combinations of preference weights. However, the model can be generalized to customize for **any combination of weights** within a continuous space during inference. Regarding more open-ended model customization (such as customizing the model based on user-provided natural language), we leave this as a direction for future work.
> > >
> > > # Q1: Guiding Principles
> > > PadiRec provides greater subjective control over the model for both users and companies, allowing preference weights to be specified arbitrarily. However, excessive autonomy may lead to more confusion. To address this,  we propose a practical approach by transforming continuous value controls into discrete click-based controls. Specifically, we could categorize preference weights into broad categories (e.g., high precision, low diversity) to make it easier for users to understand and interact.
> > >
> > >
> > >
> > > # Q2: Non-standard or Non-linear Preference Metrics.
> > > Firstly, in the **"Adapter Tuning"** step of PadiRec, the multi-objective loss function combines multiple objective functions through a linear weighted sum of the preference weights. However, in practice, we allow the use of various optimization methods (not limited to linear combinations) because our primary goal is to obtain adapters that **align with the given preference weights.** These adapters are then used to train the conditional diffusion model with adapter - preference weight pairs.
> > >
> > > Secondly, for non-standard or non-linear preferences, we can train an additional encoder to **map these preferences into the preference weight space**. This approach avoids the need for additional training of the diffusion model.

---

> > > > ### Author Response · Authors · 2024-12-03
> > > >
> > > > Dear Reviewer,
> > > >
> > > > We kindly inquire whether our responses have adequately addressed your concerns. If there are any remaining misunderstandings or uncertainties, we would greatly value the opportunity to discuss and clarify them further with you.
> > > >
> > > > Best regards,
> > > > The Authors

---

### Author Response · Authors · 2024-11-24
**Revised pdf and summary.**

Dear reviewers:

Thanks for your hard work, your suggestions really help us to improve our paper. We revised our paper according to your suggestions (**revised parts are marked as blue**) and **re-upload our modified pdf**.

We will summarize our changes as follows:

- We conducted experiments on the **SOTA backbone**, detailed in Appendix A.7.
- We supplemented the paper with an analysis of **embedding size** in SASRec, presented in Appendix A.8.
- We conducted experiments on **additional controllable objectives** in Appendix A.9 (expanding from accuracy and diversity to include fairness).
- We provided a more **detailed structure** of the diffusion model in Appendix A.13 and derived the **computation cost** for the diffusion inference process in Appendix A.14.

Finally, we emphasize that our paper primarily addresses the significant yet often overlooked challenge of inefficient retraining of deployed models at test time, thereby enhancing the models' ability to adapt dynamically to changes in task requirements. We have rigorously formulated this problem as Controllable Multi-Task Recommendation (CMTR) and, for typical recommendation systems, proposed PadiRec—a well-generalized and efficient algorithm that achieves "one instruction, one model" instead of relying on costly retraining. We kindly ask you to consider our contributions to the recommender system community, particularly in advancing the critical aspect of **controllable recommendation**.

If you have any further questions, please feel free to ask before the deadline (**Nov. 26**), and we will respond as soon as possible.

Best,

Authors

---

### Author Response · Authors · 2024-11-25
**Rebuttal deadline is approching.**

Dear Reviewers,

Thank you for your hard work in reviewing our paper and providing valuable suggestions. As the discussion deadline approaches, with less than two days remaining (**Nov. 26**), we **have not receive any responses** to our rebuttal yet. We completely understand the demands of your busy schedules, but as authors, we find ourselves anxiously awaiting your feedback.

We would greatly appreciate knowing whether our responses have adequately addressed your concerns. If you have any further questions or require clarification before the deadline, **please do not hesitate to reach out**. Your dedicated time and effort in reviewing our work are sincerely appreciated.

Best regards,

The Authors

---

### Meta-Review · Area_Chair_1xec · 2024-12-24

**Metareview:**

This paper introduces a new controllable learnign approach for multi-task recommendation, which allows the customization and adaptation of recommendation model parameters to new task requirements without retraining. The authors have performed experiments on three datasets to demonstrate the effectiveness of the proposed method. However, for this paper, the technical novelty of the proposed method seems limited. Some settings of the proposed method (e.g., predefined preference weights) also limits the application of the proposed method in real scenarios. Moreover, the experimental evaluation setting in this paper also raises some concerns. For example, the evaluation dataset Movielens-1M is a little small to validate the effectiveness of the proposed method, and Movielens dataset is also not suitable for evaluating sequential recommendation models.

**Additional Comments On Reviewer Discussion:**

In the rebuttal, the authors discussed 1) the reasonability of applying Movielens datasets for evaluating sequential recommendation models, 2) the scalability of the proposed method for handling more optimisation objectives, 3) the inference cost of the proposed method, 4) the backbone models, 5) the technical novelty of the proposed method comparing with existing works about diffusion models. The authors have provided addition experimental experiments, and addressed some concerns of the reviewers. However, the concerns regarding to the technical novelty and the experimental evaluation have not been addressed.

---

### Decision · Program_Chairs · 2025-01-22

Reject